# High-quality listening behaviors linked to social connection between strangers
Taylor N. West [1] ✉, Sara Huston[1], Kylie R. Chandler[1], Jieni Zhou[1,2] & Barbara L. Fredrickson[1]

Despite the urgent need to improve social connection, practical evidence-based recommendations on how to do so during daily interactions are lacking. One key behavior theorized to facilitate social connection is high-quality listening, yet behavioral evidence is limited. Across two pre-registered studies, we tested whether observed high-quality listening behaviors during conversations between strangers are associated with behavioral and subjective markers of social connection, and whether listening behaviors account for the effectiveness of simple interventions aimed at increasing social connection. Pairs of strangers conversed in either a 10-minute semi-structured conversation ("deep talk"; Study 1) or a brief, "small talk" opportunity (Study 2) following a randomized social connectedness intervention (total $N$ = 646). In Study 1, we found that the frequency of verbal listening behaviors (i.e., verbal validation, follow-up questions) predicted faster conversational response times and other markers of social connection (i.e., 3rd party observers and self- and partner-reports). Additionally, people randomized to a social connectedness intervention (vs. active control) asked their partner more follow-up questions (i.e., displayed high-quality listening behavior), which in turn, predicted increased social connection. We replicated and extended Study 1 to small talk conversations and found global listening behaviors also predicted behavioral and partner-reported social connection. Verbal listening indicators, however, were less consistently linked to markers of social connection and no evidence emerged that the intervention increased listening behaviors during small talk. Findings suggest observable high-quality listening behaviors may be a promising route to fostering social connection and may enhance the effectiveness of interventions aimed at improving social connection.

Interacting with strangers, for many, is a near-daily experience, offering numerous opportunities for momentary connection, or even a new relationship. Amidst the pressing need to combat loneliness[1], brief interactions outside one's inner circle of close relationships may be an overlooked but meaningful source of social connection. For instance, connecting with strangers and acquaintances is associated with a range of well-being and mental-health benefits, with effects that are independent of and comparable in magnitude to those of close relationships[2,3]. Even minimal, one-time interactions can boost well-being[4]. For instance, when people are randomized to have a genuine (vs. efficient) interaction with a barista, they experience increased positive affect and belonging[5]. Thus, moments of connection with strangers need not be long to reap benefits, just heartfelt.

Despite the positive benefits of interactions with strangers, people often report avoiding doing so alongside exaggerated fears regarding their own lack of conversational skills or enjoyment[6]. These findings, in conjunction with the ongoing "loneliness epidemic"[1], suggest people need simple, actionable strategies to improve connection in their daily interactions, yet evidence-based behavioral recommendations are limited. One key behavior theorized to facilitate social connection (with any interaction partner) is high-quality listening[7,8]. High-quality episodic listening is defined as a multi-faceted, holistic process with three key components: attention, comprehension and positive intention[9]. High-quality listening is communicated to speakers via a range of observable verbal and nonverbal behaviors, such as follow-up questions, verbal validation, eye gaze, facial expressions, back-channeling utterances (e.g., mhmm), and body postures[9]. Based on Episodic Listening Theory (ELT)[9], a listener's internal, unobservable behaviors (e.g., attention, comprehension) influence their external, observable behaviors (e.g., eye gaze, nodding, question-asking). These observable listening behaviors, in turn, shape the speaker's perception of how well they are being listened to. Cues of high-quality listening may be the initial signal to speakers

[1]Department of Psychology and Neuroscience, University of North Carolina at Chapel Hill, Chapel Hill, NC, USA. [2]Department of Psychology, Yale University, New Haven, CT, USA. ✉e-mail: taywest@live.unc.edu

of a listener's positive intentions and openness, thus setting the stage for meaningful connection[10]. Because strangers have few cues to predict one another's intentions beyond immediate, observable behaviors, those indicative of high-quality listening may be vital for fostering high-quality connection.

For interpersonal benefits to emerge, one interactant's listening is insufficient if the speaker does not notice it. The speaker must *feel* listened to. Accordingly, a partner's observable listening behaviors are known to drive a speaker's perceptions of being listened to[9,11]. Amongst the varied behavioral cues a listener may display, explicit verbal expressions (e.g., paraphrasing the speaker, expressing empathy) have been argued to be the strongest signal that conveys high-quality listening to a speaker, as they can provide direct evidence of having understood the speaker[12,13]. In contrast, weaker evidence, such as backchannel responses (e.g., short verbal responses such as "yeah," "uh-huh") and/or nonverbal listening cues (e.g., nodding) can be deceptively used when people merely pretend to listen[13]. Consistent with this, several studies have shown that verbal markers of supportive listening were stronger predictors than nonverbal markers of positive outcomes following a speaker's disclosure of an upsetting event to a trained confederate[14,15]. Given the heightened uncertainty characteristic of interactions between unfamiliar partners, explicit verbal indicators may be especially useful for building connection by helping interactants better anticipate one another's thoughts and positive intentions.

Historically, research on listening has been conducted in clinical (e.g., counseling[16]), marital, and organizational (e.g., managerial communication, sales) contexts. These settings do not adequately capture the dynamics of informal social interactions outside close relationships or hierarchical power dynamics, and it remains unclear to what extent findings generalize outside such interactions. Thus, listening behaviors in informal social contexts are surprisingly understudied. Recently, however, social psychologists have paid growing attention to the concept of listening as important to social interactions[9,17]. A recent meta-analysis of the workplace outcomes associated with listening quality (i.e., affect, cognition, relationship quality, performance) revealed that perceived listening during workplace conversations was most strongly positively related to relationship quality (total $N = 108,185$)[18]. Moreover, a 3-week listening training program produced greater feelings of relatedness with colleagues compared to an active control group[11]. Outside organizational contexts, in a one-time laboratory study, participants randomized to disclose an experience of past social rejection to a stranger who engaged in high (vs. low) quality listening, reported greater feelings of connection, which in turn predicted decreased loneliness[17]. This emerging evidence that high-quality listening may foster high-quality social connection has spurred a recent wave of scholars hypothesizing links between high-quality listening and constructs closely associated with high-quality social connection, such as perceived partner responsiveness[7,19,20], conversational response time[21], and perceived safety and positivity resonance[8]. However, limited empirical evidence directly supports these posited links, and research is especially needed to investigate listening behaviors that are merely observed and coded systematically (i.e., in comparison to manipulated or captured via the speaker's reported perceptions) or behavioral markers of social connection.

To assess social connection, we triangulated across diverse behavioral and self-reported measures, drawn from multiple theoretical perspectives, to ensure a valid and robust evaluation. First, we assessed conversational response time, the speed in milliseconds, precluding conscious awareness, at which people respond to each other in conversation, which has been empirically linked with feelings of connection and enjoyment, both as reported by interactants and observers[22]. Perhaps counterintuitively, we propose that quick responding often hinges on careful listening, yet research is needed to empirically test this link. Instead, some scholars have argued that listening can also be measured by response times, thereby conflating listening with social connection[21]. Critically, however, Episodic Listening Theory (ELT) postulates that high-quality listening *precedes* social connection, as listening, an individual act with cognitive and behavioral components, gives rise to an emergent collective experience of connection or

togetherness[9]. Observed listening behaviors (i.e., listeners' conversational response *content*) may determine speakers' perceptions of having been heard and, in turn, speakers' perceptions of connection and their own faster conversational response time. When a speaker perceives a listener to be actively engaged, they may need less cognitive effort to monitor the listener's reactions or puzzle over how to proceed. Thus, when strangers converse, observed listening behaviors may prompt faster conversational responses that, in turn, signal that partners are "in sync" with one another.

Second, we assessed a marker of mutually experienced social connectedness termed positivity resonance, which is theorized to be akin to the dyadic state of togetherness postulated to arise from listening by ELT[9]. Positivity resonance is a collective positive emotional state marked by mutual kind-heartedness and nonverbal and biological synchrony[23] and is measurable by self-reported perceptions[24] and behavioral indicators[25]. Zhou and Fredrickson[8] speculated that high-quality listening fosters the conducive conditions of perceived safety and real-time sensory connection and thereby sets the stage for positivity resonance (a.k.a., togetherness) to emerge. Recent experimental work has established a causal link from high-quality listening to perceived positivity resonance. Across four studies in which strangers conversed about disagreements, participants randomized to a high-quality listening condition (versus low and/or moderate) reported the greatest perceived positivity resonance[26]. Additionally, when listeners were randomized to engage in high-quality listening (versus moderate quality) during a discussion of character strengths, both the speaker and listener reported greater perceived positivity resonance[27]. However, because this past work manipulates the quality of listening behaviors, research is needed to test whether observed individual-level displays of listening behaviors during casual or getting-acquainted conversations are also associated with self- and partner-reported perceived positivity resonance, as well as behavioral indicators of positivity resonance.

In the present research, we sought to examine the link between high-quality listening *behaviors* and social connection, while also raising the rigor of assessments of high-quality social connection. Incorporating behavioral data provides a more objective measure than self-report and enables the identification of specific, actionable behaviors that individuals can adopt to effectively signal high-quality listening to their interaction partners. We draw on two secondary datasets, each of which consisted of an intervention aimed at improving quality social connection (i.e., positivity resonance), followed by either semi-structured "deep talk" (Study 1) or small talk (Study 2) with a stranger. For logistical and experimental control, albeit at the cost of potential practice effects, each participant's partner was either a trained confederate (Study 1) or experimenter (Study 2), always blinded to the participant's randomized condition. Given the theory that high-quality listening precedes social connection[8], we had the opportunity to additionally examine whether high-quality listening behaviors during post-intervention interactions emerge as a potential mechanism through which each intervention successfully promoted social connection. Each study previously reported evidence that an intervention that encouraged participants to have more high-quality connections (compared to controls) was successful. Specifically, the intervention of Study 1 successfully raised markers of social connection during an in-lab interaction 48 h later[2] and that of Study 2 improved self-reported positivity resonance in daily interactions across 35 days[28]. To the extent that listening behaviors are robustly associated with markers of social connection, we expect that behavioral interventions designed to improve social connectedness may implicitly draw out higher-quality listening behaviors. That is, as individuals seek and seize opportunities to connect, they may choose, consciously or unconsciously, to enact high-quality listening. By indicating attentiveness and positive intention, behaviors indicative of high-quality listening may thus function as an active ingredient responsible for the success of social connectedness interventions.

We preregistered a similar set of hypotheses across both studies. First, we hypothesized (Hypothesis 1) that participants' higher-quality listening behaviors would be associated with higher-quality social connection as indicated by faster conversational response time (negative association) and greater positivity resonance (i.e., self-reported, partner-reported, and

behaviorally coded; positive association). Next, we hypothesized (Hypothesis 2) that a behavioral intervention aimed at improving social connection would also increase participants' high-quality listening behaviors during in-lab conversations. To the extent that these interventions increased listening behaviors, we explored whether they, in turn, successfully improved social connection, as mediated by enhanced listening behaviors (Exploratory Hypothesis).

## Methods
### Study 1
In a first test of our two preregistered hypotheses, we focused on a context in which associations between high-quality listening and high-quality social connection may be most likely to emerge: during semi-structured conversations designed to facilitate interpersonal closeness. We operationalized listening behaviors in Study 1 with a focus on two types of explicit verbal expressions in conversation that reflect attention, comprehension, and positive intention: verbal validation and follow-up questions (see Supplementary Section I for a discussion of why we elected to forego preregistered coding for nonverbal listening indicators).

### Participants and procedure
Participants ($N = 335$) were undergraduate and graduate students originally recruited from the University of North Carolina at Chapel Hill as part of a larger study on "Technology and Behavior Goals"[2] between Fall 2021 and Winter 2022. This secondary analysis draws only on participants who attended the post-intervention lab session and had video and/or audio recordings available ($N = 300$; social connectedness $n = 157$ or active control $n = 143$). Participants in the analyzed sample were primarily women (women $n = 225$; men $n = 66$; other $n = 4$) between the ages of 18 and 35 ($M = 20.02$, $SD = 2.63$), and identified as either White (50%), Black (4.7%), Asian (25.5%) or Hispanic (10.4%), with the remaining 9.4% identifying as either multiracial, other, or preferred not to say. Participants were compensated up to $25 or course credit, with up to $5 in bonus payment. Sample size was originally determined based on a power analysis for the original aims of the grant-supported study. A sensitivity power analysis was conducted in G*power 3.1[29] for a linear multiple regression with 1 primary predictor and 4 covariates ($N = 300$, $\alpha = 0.05$ (two-tailed), power = 0.80). Primary models are powered to detect a minimum effect size of $f^2 = 0.03$ (small).

The Institutional Review Board at UNC-Chapel Hill approved all study procedures (IRB #19-3258). Hypotheses and analyses for the current study were preregistered on May 17, 2023, at https://osf.io/nh9j6?view_only= 9a4c08436c6d41e3b639c6d67c6c55ce. Eligible participants provided informed consent and attended two 30-min in-lab sessions approximately 48 h apart. During the first lab session, participants were randomized to either a stranger and weak-tie (i.e., acquaintances or other non-close others) social connectedness intervention condition or a diaphragmatic breathing intervention (active control group). This active control group was selected to be a simple non-social health behavior that participants could frequently enact across 24 h, to parallel the cognitive and motivational effort of the treatment group[2]. In both conditions, the intervention consisted of a 3-min educational video followed by a conversation with a virtual human avatar called "Ellie." Participants randomized to the social connectedness condition watched a video about the benefits of positive connections with strangers and acquaintances (https://www.youtube.com/watch?v= tXTj4mDON7k) and were subsequently encouraged by Ellie to be more attentive and open to connection with strangers and acquaintances. Participants randomized to the diaphragmatic breathing condition watched a video about the benefits of using proper breathing techniques (https://www. youtube.com/watch?v=27_Z-zaFb88) and were subsequently encouraged by Ellie to focus on allowing their breaths to expand from their back and belly. Ellie was animated using the "Wizard of Oz" method in which a hidden, trained research assistant controlled Ellie remotely using a predefined script. Within each intervention condition, participants either interacted with Ellie with additional avatar nonverbal cue features enabled

or disabled (e.g., smiles and head nods; given limited findings for this second randomized variable in the original report[2], the nonverbal cue condition was controlled for and not analyzed. See Supplementary Section I for pre-registered exploratory analyses). In both conditions, Ellie asked participants to create behavioral implementation intentions for their assigned behavioral goal (i.e., If-Then plans[30]; see Supplementary Material Section II for the full Ellie script). Participants were instructed to carry out their goal of either having more moments of quality connection with strangers and weak ties or practicing diaphragmatic breathing until their second lab session.

Approximately 48 h after the first lab session, participants returned for a second 30-min lab session in which they participated in a 10-min audio and video-recorded fast friends task with a stranger[31]. Two partially concealed cameras, one facing each interaction partner, were used to record the conversation. The "stranger" was always one of ten trained female confederates, blind to the randomized condition. The task consisted of a stack of printed questions, ordered by increasing levels of self-disclosure. The confederate always read and answered the same first question, with a standardized response, to set the tone and level of intimacy. After the first standardized response, confederates were trained to match the emotional intensity and level of intimacy of the participant. Following the 10-min conversation, both the participant and confederate were moved to separate rooms to report on the perceived positivity resonance during the interaction. Following the conversation, participants provided consent for their audio/ video data to be analyzed. Those who indicated audio-only consent ($n = 14$) were not behaviorally coded. Participants were debriefed at the end of the study.

**Behavioral indicators of high-quality listening**. Drawing from Kluger and Itzchakov's[9] definition of high-quality listening, we developed a behavioral coding scheme to capture participants' high-quality listening during the fast friends task[31]. The coding system operationalized listening based on two verbal cues that demonstrate attention, comprehension, and positive intention: asking relevant follow-up questions and verbal validation. Follow-up questions were defined as questions in which the listener conveys wanting to learn more, better understand, or clarify something the speaker said (coders excluded the initial questions asked in the fast friends prompts). Verbal validation was defined as expressing understanding, paraphrasing, or reflecting information the listeners heard, or as offering support or a friendly opinion on something the speaker said. This did not include short backchanneling indicators (e.g., yeah, mhmm) or swiftly turning the conversation back to the self by expressing simple agreement (e.g., "me too"), rather than first acknowledging or reflecting the speaker's statement (Full instructions to coders are provided in the Supplementary Section III). The coding team consisted of 5 trained research assistants. For each 30-s bin of the video-recorded interaction, coders recorded the frequency of follow-up questions and verbal validation (separately) enacted by the participant. Coders were instructed to review each bin twice. Coders were trained over three weeks, with 3 sets of 5 practice-coding videos. Following each set, the team met to discuss coding disagreements. Once reliability was met (i.e., threshold ICC > 0.80) in the training phase, two coders independently coded a set of videos each week for six weeks. To assess bin-by-bin inter-rater reliability, a set of 20% of the videos was coded by all 5 coders, spread across the coding period (Follow-up questions ICC = 0.97; verbal validation ICC = 0.94). Weekly meetings continued throughout the coding period for coders to discuss disagreements for the purposes of maintaining reliability. Discussed disagreements in codes were not changed. The average frequency of follow-up questions and verbal validation across coders was used for analysis.

**Conversational response time**. Response times were computed to the 100th ms using the automated transcription service AssemblyAI (assembleyai.com). Following previous work[22,32], conversational response time was calculated by subtracting the end timestamp of the last speaker's turn from the start timestamp of the current speaker's turn,

then computing the average response time (i.e., gap between speech) for each speaker across the conversation. Research assistants manually checked transcriptions for completeness and accuracy and flagged unusable files with major errors, primarily due to poor audio quality that resulted in missing or unreliable auto-transcription. Of the data used in the present study, 12 transcript files were flagged as unusable and were not included in response time analyses (we also note that the original preregistration for Study 1 specified hypotheses only for participant response time and did not include partner response time).

**Self-reported perceived positivity resonance.** Participants and confederates individually completed an abbreviated 3-item Perceived Positivity Resonance Scale[24]. Specifically, participants (and confederates) were instructed: "Thinking about the conversation you just had, please move the slider to reflect your estimate (from 0% to 100%) of how much of the time during this interaction did you… (1) experience a mutual sense of warmth and concern toward one another? (2) feel a mutual sense of being energized and uplifted in each other's company? (3) feel 'in sync' with the other person?" These three items were averaged together. Cronbach's alpha was 0.87 for participants and 0.93 for confederates.

**Behavioral indicators of positivity resonance (BIPR).** We closely followed the behavioral coding system originally developed for married couples[25], slightly adapted for stranger dyads (e.g., coding for positive affirmation instead of using terms of endearment)[33]. Behavioral indicators of positivity resonance, or BIPR, were assessed on an intensity scale of 0 (not present), 1 (low intensity), 2 (high intensity) for each 30-s bin across the 10-min video using the following prompt: "Did positivity resonate between the two partners? That is, did they show actions, words, or voice intonation that conveyed mutual warmth, mutual concern, mutual affection and/or shared tempo (i.e., shared smiles and laughter)?" The low intensity was defined as a bin that included one instance of shared laughter between the interaction partners or an instance of affirmations that conveyed positive affect. High intensity was defined as either two or more shared smiles or laughter (or one instance that lasted half the bin) or two or more instances of positive affirmation. The coding team consisted of six trained research assistants split into two groups, with each group coding half the videos. 20% of videos were coded by all six research assistants to establish reliability (ICC = 0.73). Scores on behavioral indicators of positivity resonance reflected the total sum of the average score across coders for each bin (Three of the six research assistants on the BIPR coding team were also on the listening coding team. However, training and coding for BIPR were completed before training and coding for listening began).

**Analysis plan**
All analyses were conducted in R (4.3.1). We first inspected variables and model residuals for normality in preliminary analyses. All coded variables (verbal listening behaviors and BIPR), and the conversational response time variables were highly positively skewed with QQ-plots suggesting non-normally distributed residuals. To correct for this, we took two approaches. First, to minimize the effect of a few extreme outliers, we winsorized values falling over +3 SD. Next, we log-transformed variables. Visual inspection of the resulting QQ-plots suggested improvements in the distribution of residuals. In the Supplementary materials, we additionally report all model results using raw variables (i.e., neither winsorized nor transformed; Supplementary Tables S1.5–S1.7). Both self-reported positivity resonance variables (participant and confederate) were rescaled from the original 100-point scale by dividing by 10 for analysis. Hypotheses were tested using multiple regression and multi-level models. All tests were two-tailed. For testing associations in Hypothesis 1, listening variables were entered as predictors (separately) and social connection variables were entered as dependent variables (i.e., participant conversational response time, participant-reported positivity resonance, and BIPR). Because confederates had repeated interactions, to account for non-independence of

observations, we tested effects of listening predictors on partner conversational response time and partner-reported positivity resonance using multilevel models (using the package lmerTest[34]) with confederate identity as a random intercept (We note that this deviates from our preregistered analysis, which specified multiple regression to test all hypotheses. However, this would fail to account for repeated observations and is thus the incorrect statistical approach). For testing condition effects in Hypothesis 2, listening variables were entered as individual outcomes. All models additionally controlled for intervention condition, nonverbal cue condition, and participant gender. To minimize the effect of individual differences across confederates, we additionally included confederate identity as a categorical variable in all analyses. Unadjusted models are reported in the supplementary materials (Supplementary Tables S1.1–S1.4).

For listening behaviors that were significantly increased by the social connectedness intervention, we proceeded with an exploratory mediation model in which the randomized condition predicted a latent factor of social connection via listening behaviors (Exploratory Hypothesis). For the latent factor, we conducted a confirmatory factor analysis (CFA) using the lavaan package[35] in R, loading all previously tested social connection indicators (i.e., participant and partner conversational response times, participant- and partner-reported positivity resonance, and BIPR). Model fit was assessed using a variety of established indicators, including the comparative fit index (CFI), the root mean square error of approximation (RMSEA) and the standardized root mean square residual (SRMR), using recommended cutoffs (CFI > 0.95, RMSEA < 0.08, SRMR < 0.08[36]). Mediation model parameter estimates were obtained using a full information maximum likelihood estimator and unbiased standard errors using bootstrapping with 1000 resamples. We additionally controlled for nonverbal cue condition, participant gender and confederate effects in the reported exploratory model. To test whether the hypothesized temporal order of the exploratory model best fit the data, we additionally tested a model in which we swapped the mediator and outcome such that the latent factor of social connection (alternative mediator) explained the intervention effects on listening behaviors (alternative outcome). This alternative model followed the same steps and guidelines explained above.

**Study 2**
In Study 1, we tested hypotheses in a context designed to facilitate closeness between strangers. Yet, everyday interactions between strangers typically do not involve semi-structured, intimate conversation but instead brief, polite conversation on mundane, non-controversial topics. Furthermore, we sought to expand on the verbal indicators of high-quality listening used in Study 1 to also test a holistic measure of global listening behaviors that considers nonverbal behaviors together with explicit verbal expressions. Although explicit verbal expressions may provide the strongest "honest signal" of high-quality listening, the extent to which nonverbal behaviors are also present may serve to further amplify this signal. In Study 2 (N = 348), raising the bar to a small talk context, we tested whether high-quality listening behaviors, measured both as verbal indicators (as in Study 1) and as a global, holistic evaluation of verbal and nonverbal indicators, are linked to high-quality social connection in a brief conversation with an unfamiliar partner. In this study, participants were initially randomized to either a passive control group (n = 89) or one of three intervention conditions: social connectedness in general (n = 85), social connectedness with strangers and acquaintances (n = 86), or mindfulness (n = 88). After 35 days of daily reporting and practice implementing their behavioral goal, participants attended an in-lab session in which an opportunity for small talk between the experimenter and participant was surreptitiously arranged.

First, we tested a Validation Hypothesis, that during small talk, a new global behavioral coding scheme that captures a holistic synthesis of verbal and nonverbal components would be positively correlated with the verbal listening codes used in Study 1. All other hypotheses pattern those of Study 1. Specifically, we tested Hypothesis 1, which states that participants' listening behaviors (verbal and global) would be negatively associated with both their own and the experimenter's conversational response time, and

positively associated with both partner-reported and behaviorally coded positivity resonance. Next, we tested Hypothesis 2, which states that social connectedness behavioral interventions would increase listening behaviors. Given results from the original research report on this data showed that mindfulness (the active control) increased perceived positivity resonance to a comparable degree as the two social connectedness interventions[28], consistent with other published secondary analyses of these same data[37], we grouped the three active conditions together to compare against the passive control group (Although consistent with past research, this approach deviates from our preregistered hypothesis to test only social connectedness interventions, results of which can be found in reported individual condition effects in Supplementary Table S2.3).

## Participants and procedure

Participants ($N = 416$) were recruited from the University of North Carolina Chapel Hill to participate in a larger study ("Daily Wellness Study"[28]) between March and November of 2019. This secondary analysis draws only on participants who attended the lab session at the end of a 35-day intervention and had video/audio data from the session available. This left us with an analyzed sample of $N = 348$ ($M_{age} = 34.02$, $SD_{age} = 11.28$). The sample was primarily women (women $n = 282$; men $n = 62$), with a majority racially identifying as White (69.8%), followed by Black or African American (11.6%), Asian (8.2%), Hispanic or Latin American (7.3%), or other (3.1%). To be eligible, participants must have been between the ages of 20 and 65, working part- or full-time, not currently enrolled as an undergraduate student, and have access to a computer/mobile phone. Participants could earn up to $100 for participation, plus the chance to win an additional $100 in a raffle based on the number of surveys completed over 35 days. Sample size was determined based on a power analysis for the original aims of the grant-supported study. A sensitivity power analysis was conducted in G*power 3.1[29] for a linear multiple regression with 1 primary predictor and 4 covariates ($N = 348$, $\alpha = 0.05$ (two-tailed), power = 0.80). Primary models are powered to detect a minimum effect size of $f^2 = 0.023$ (small).

The Institutional Review Board at UNC-Chapel Hill approved all study procedures (IRB #18-2810). Hypotheses and analyses for the current study were preregistered on October 14, 2023, at https://osf.io/xz39m/?view_only=c07943eee8a14dddb159e1b43c085405. After confirming eligibility, participants provided informed consent and completed an online pre-intervention survey, 35 nightly online self-reports, a post-intervention survey and one in-person post-intervention lab session. Here, we present results only on video/audio and partner-reported data from the post-intervention lab session, which have not been reported elsewhere. Participants were randomized to one of four conditions. Two connectedness conditions watched the same 11-min video on the importance of positive connections with others (https://www.youtube.com/watch?v=fHoEWUTYnSo). Following the video, one connectedness group was instructed to try and seek more frequent positive connections with people in general ("social connectedness–general"), while the other was instructed to seek more frequent positive connections outside their close circle of friends and family ("social connectedness—weak ties"). The third intervention group viewed a 9-min video on the importance of mindful awareness (https://www.youtube.com/watch?v=qzR62JJCMBQ) and was instructed to try to experience more frequent mindfulness in daily life. All three intervention groups received daily reminders via email of their condition-specific behavioral goal. The fourth condition served as a monitoring passive control group, and received nightly surveys, but no intervention or reminders.

At the end of the 35-day intervention period, participants visited the lab for a final in-person session led by one of four experimenters. Experimenters remained blind to participant condition. After participants provided consent, the experimenter surreptitiously created an opportunity to engage in small talk with the study participant. To do so, the experimenter pretended they were unable to access the study laptop and had to wait for a password via text from a team member. At this time, rather than email their team member, the experimenter used the "new message" app to schedule a fake text to arrive in 5 min. The experimenter was instructed to attempt to strike up a conversation with the participant, asking as many of the following questions as needed, in the order presented here: To start, "So, how does the rest of your day look?", then, as a good news script, "I'm really looking forward to the end of the day—I got a call before the session and found out that I can pick up the keys to my new apartment and move in this weekend. It's a bit earlier than I expected, so I'm excited now; I've only ever lived with roommates and it's my first place by myself," and "You doing anything fun this weekend?" The experimenter was instructed to allow the participant to carry the conversation if they would or continue with the next question. After 5 min, or if the conversation ceased (whichever happened first), the experimenter checked their phone and entered the "received" password to resume the study procedures. One camera, which captured a side view of the dyad, audio and video recorded this portion of the in-lab session. At the end of the in-lab session, participants were debriefed and asked to provide video and audio consent for the recorded portion of their visit.

**Behavioral indicators of high-quality listening.** We assessed follow-up questions and verbal validation, following the same procedures as described in Study 1. To assess inter-rater reliability, a team of 5 trained research assistants coded 20% of the videos across the coding period (follow-up questions ICC = 0.96; verbal validation ICC = 0.92). To assess global listening behaviors (inclusive of verbal and nonverbal cues), we trained a new, independent team of 3 research assistants (not previously trained or familiar with the verbal indicators scheme used here or in Study 1). Continuing to draw on Kluger and Itzchakov's[9] definition, we modeled the coding scheme after the BIPR instructions. Specifically, for each 30-s bin, coders responded to the following prompt: "Did the participant exhibit high-quality listening?" That is, did they show attention (when in the listening role) via backchannel behaviors (verbal and nonverbal), comprehension through paraphrasing and asking open questions, or convey positive intention? Responses were on a 5-point scale based on the intensity, duration and clarity of behaviors, with 0 = none or very brief, 1 = a little bit (a few times or at low intensity), 2 = a moderate amount (e.g., one or more behaviors displayed, maybe half the time or something with low intensity), 3 = more than a moderate amount, 4 = a lot (e.g., multiple behaviors displayed for a majority of the time and at high intensity). The full team of 3 coded all videos and achieved high reliability (ICC = 0.87).

**Conversational response time.** We calculated conversational response time using the same methods and computations as in Study 1.

**Perceived positivity resonance.** Due to the mild deception used to stage an opportunity for small talk with the experimenter, self-reported positivity resonance was not collected from participants. Only the experimenter reported on perceived positivity resonance following the interaction. This study used the original 7-item Perceived Positivity Resonance Scale[24], using the same 100-point sliding scale as in Study 1. Cronbach's alpha was 0.98.

**Behavioral indicators of positivity resonance.** We followed the procedures for the original coding scheme[25]. As these videos were coded for BIPR four years prior to Study 1's BIPR coding, adjustments made to the coding scheme reported in Study 1 (i.e., coding affirmations) were not applied to Study 2. A team of four trained research assistants was randomly assigned videos, with two coders per video. No members of the BIPR coding team participated in either of the listening coding teams. 20% of videos were coded by all trained coders for assessing reliability (ICC = 0.89). Positivity resonance scores were computed by taking the average sum score across coders.

## Analysis plan

Analyses followed closely those used in Study 1. Upon inspection of variables for normality, only verbal listening behaviors and conversational response times were positively skewed. In this study, BIPR and global listening behavior codes followed normal distributions. For positively skewed

variables, patterning Study 1, we winsorized and log-transformed variables. The Supplementary material reports all models with raw variables (neither winsorized nor transformed). Because interaction videos varied in length following the natural course of small talk (5 min or less), we further divided the total sum score of BIPR and verbal listening variables by the total number of 30-s bins scored. Thus, scores reflect per-bin average sums across coders. Because global listening behaviors were not assessed as a sum frequency, we did not need to compute per-bin sums. That is, global listening behaviors already reflect the average score across bins. To control for the length of the interaction, the number of bins coded was entered as a covariate in models that include this non-count variable. Experimenter perceived positivity resonance was rescaled by dividing by 10 for analysis, again patterning Study 1. To test the Validation Hypothesis, we conducted partial correlations between listening behaviors (i.e., global, verbal validation, follow-up questions), controlling for condition with pair-wise deletion. Confidence intervals for the partial correlations were estimated using nonparametric bootstrapping (2000 resamples) with the bias-corrected and accelerated method (BCa). We conducted multiple regressions and multi-level models to test Hypothesis 1 and multiple regressions to test Hypothesis 2 (As in Study 1, multi-level analyses were not originally preregistered to test experimenter outcomes but is the correct statistical approach to account for repeated observations). All tests were two-tailed. For testing associations (Hypothesis 1), listening variables (i.e., verbal validation, follow-up questions, global) were entered as predictors in separate models, and social connection variables were entered as separate dependent variables (i.e., participant conversational response time and BIPR). For the dependent variables of partner conversational response time and partner-reported positivity resonance, as in Study 1 we tested effects of listening predictors using multi-level models (using the package lmerTest[34]) with experimenter identity as a random intercept to account for non-independence of observations. For testing condition effects on listening (Hypothesis 2), the randomized condition was recoded with treatment groups combined (social connectedness–general, social connectedness—weak ties, mindfulness = 1) compared to the passive control group (= 0). As in Study 1, all reported models controlled for condition, participant gender and experimenter effects with unadjusted models reported in the Supplementary materials.

## Reporting summary

Further information on research design is available in the Nature Portfolio Reporting Summary linked to this article.

## Results

### Study 1

Descriptives of study variables are presented in Table 1.

**Hypothesis 1: Associations between listening behaviors and markers of social connection. Conversational response time**: Supporting our hypothesis, participants who asked more follow-up questions had a faster conversational response time and interacted with partners with a faster response time (participant: $\beta = -0.23$, $b = -0.10$, 95% CI [−0.16, −0.05], $p < 0.001$; partner: $\beta = -0.24$, $b = -0.08$, 95% CI [−0.13, −0.04], $p < 0.001$; see Fig. 1A). Likewise, those who engaged in more verbal validation also had a faster conversational response time or interacted

**Table 1 | Study 1 descriptive statistics for primary study variables**

| Variable | M (SD) | N | Scale |
|---|---|---|---|
| 1. Avg gap length—participant | 1.01 (0.79) | 286 | Seconds |
| 2. Avg gap length—partner | 0.86 (0.65) | 286 | Seconds |
| 3. Participant-reported PosRes | 7.87 (1.53) | 300 | 0–10 |
| 4. Partner-reported PosRes | 7.91 (1.67) | 281 | 0–10 |
| **Behavioral codes per 30 s bin** | | | |
| 5. BIPR | 0.39 (0.32) | 286 | 0–2 |
| 6. Verbal validation | 0.72 (0.59) | 286 | Avg frequency |
| 7. Follow-up questions | 0.19 (0.28) | 286 | Avg frequency |

*M* and *SD* are used to represent mean and standard deviation, respectively.
We report per-bin averages (by dividing by 10) to allow for comparison to Study 2 descriptives. Avg frequency refers to the average count frequency across coders per bin.
*PosRes* positivity resonance, *BIPR* behavioral indicators of positivity resonance.

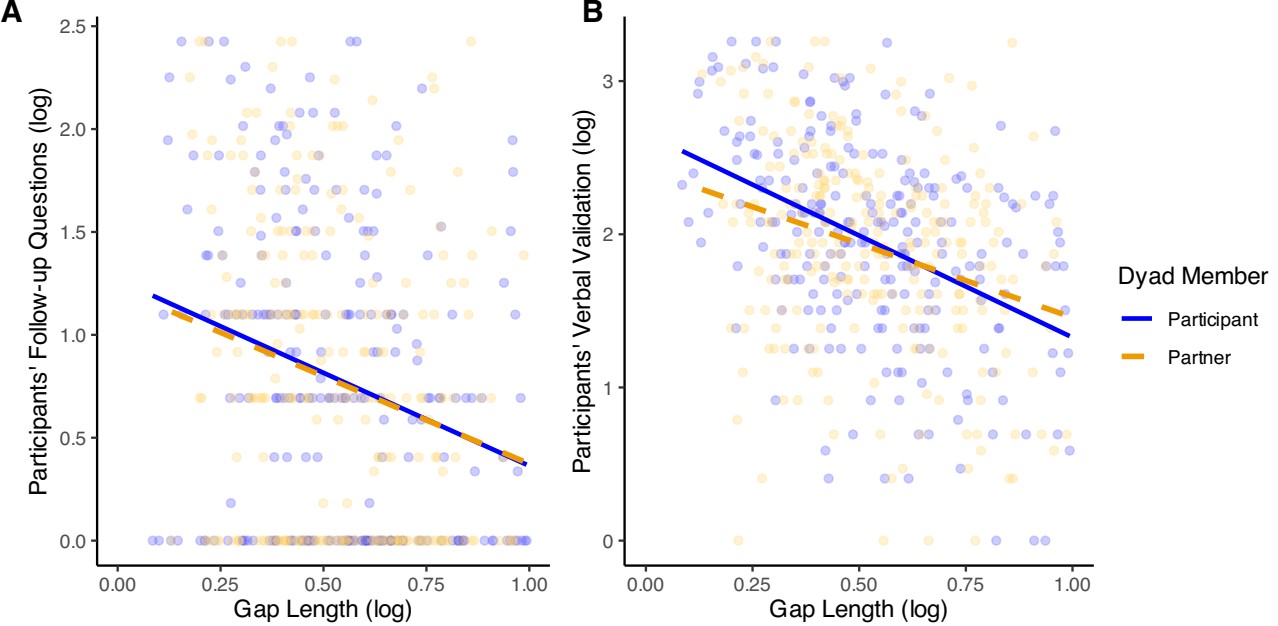

**Fig. 1 | Study 1: Participants' listening behaviors and conversational response time.** *N* = 272. Figures display results of models testing participants' follow-up questions (**A**, left) and verbal validation (**B**, right) on participant and partner conversational response times.

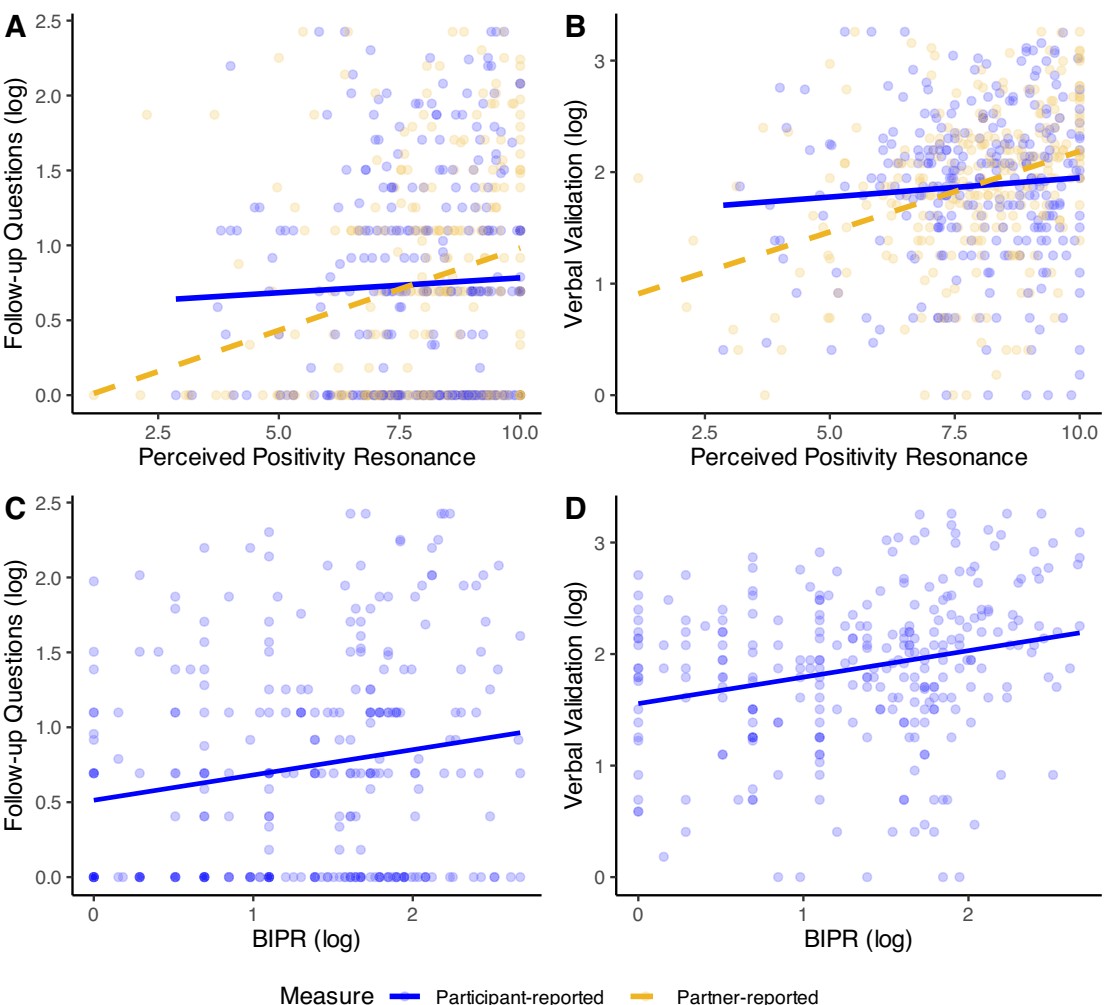

**Fig. 2 | Study 1: Participants' listening behaviors and positivity resonance.** Figures display results of models testing participants' follow-up questions (**A**, top left, participant $N = 284$, partner $N = 267$) and verbal validation (**B**, top right, participant $N = 284$, partner $N = 267$) on participant and partner-reported perceived positivity resonance, as well as those testing participants' follow-up questions (**C**, bottom left, $N = 286$) and verbal validation (**D**, bottom right, $N = 286$) on behavioral indicators of positivity resonance (BIPR).

**Table 2 | Study 1 results of participants' listening behaviors on positivity resonance**

| Model | Listening behavior | *B* | b | 95% CI | *p*-value |
|---|---|---|---|---|---|
| Participant-reported Positivity resonance | Follow-up questions | 0.05 | 0.10 | −0.15, 0.35 | 0.417 |
| | Verbal validation | −0.00 | −0.00 | −0.25, 0.25 | 0.999 |
| Partner-reported positivity resonance | Follow-up questions | 0.22 | 0.49 | 0.23, 0.76 | <0.001 |
| | Verbal validation | 0.33 | 0.80 | 0.53, 1.06 | <0.001 |
| BIPR | Follow-up questions | 0.13 | 0.12 | 0.01, 0.23 | 0.03 |
| | Verbal validation | 0.20 | 0.21 | 0.10, 0.32 | <0.001 |

$N = 300$.
Standardized betas presented. Unstandardized betas and associated 95% confidence intervals are also presented.
*BIPR* behavioral indicators of positivity resonance.

with partners with faster response times (participant: $\beta = -0.42$, $b = -0.20$, 95% CI [−0.25, −0.15], $p < 0.001$; partner: $\beta = -0.31$, $b = -0.11$, 95% CI [−0.16, −0.07], $p < 0.001$; see Fig. 1B). The pattern of results was unchanged in unadjusted models (see Supplementary Tables S1.1, S1.2).

**Positivity resonance**: Contrary to our hypothesis, we did not find evidence that either listening behavior was significantly associated with participant-reported positivity resonance (see Fig. 2A, B, blue line). However, supporting our hypothesis, greater observed follow-up questions and

verbal validation were each positively associated with partner-reported and behaviorally coded positivity resonance during the 10-min interaction (see Table 2 for results summary; see Fig. 2A–D). The pattern of results was unchanged in unadjusted models (see Supplementary Table S1.3).

**Hypothesis 2: Effect of social connectedness intervention on listening behaviors. Social connectedness intervention effects**: Supporting our hypothesis, participants randomized to connect with strangers and acquaintances, compared to an active control group, asked

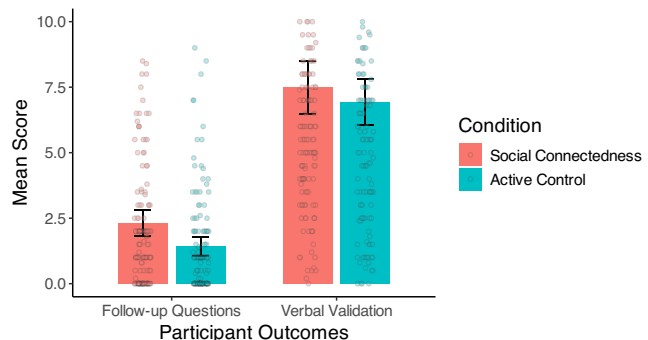

**Fig. 3 | Study 1: Social connectedness intervention effects on participants' listening behaviors.** $N = 286$. This bar chart displays the distribution and average frequency of participants' follow-up questions and verbal validation for each condition (social connectedness $n = 147$; active control $n = 139$) using the raw variables. The significance level indicated is based on the reported models. $**p < 0.01$

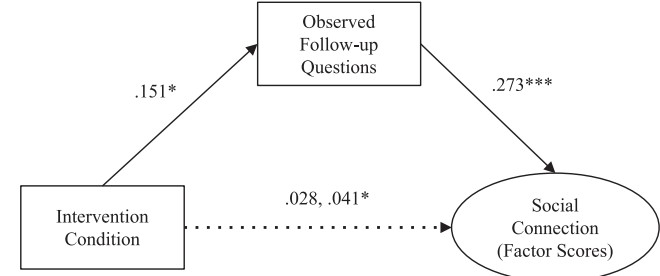

**Fig. 4 |** $N = 271$. Exploratory mediation model for the effect of the social connectedness intervention on the quality of social connection via participants' follow-up questions. Standardized coefficients presented. $*p < 0.05$, $***p < 0.001$.

**Table 3 | Study 2 descriptive statistics for primary study variables**

| Variable | M (SD) | N | Scale |
|---|---|---|---|
| 1. Avg gap length—participant | 0.43 (0.30) | 303 | Seconds |
| 2. Avg gap length—partner | 0.44 (0.30) | 303 | Seconds |
| 3. Partner-reported PosRes | 8.08 (1.71) | 348 | 0–10 |
| **Behavioral codes per 30 s bin** | | | |
| 4. BIPR | 0.58 (0.33) | 348 | 0–2 |
| 5. Global listening | 1 (0.55) | 347 | 0–4 |
| 6. Verbal validation | 1.10 (0.72) | 348 | Count |
| 7. Follow-up questions | 0.32 (0.26) | 348 | Count |

$M$ and $SD$ are used to represent mean and standard deviation, respectively.
$PosRes$ positivity resonance, $BIPR$ behavioral indicators of positivity resonance.

more follow-up questions during the structured conversation. This was a small but statistically significant effect ($\beta = 0.15$, $b = 0.21$, 95% CI [0.04, 0.26], $p = 0.009$), corresponding to a 23% increase in the number of follow-up questions. On the raw (non-transformed) scale, the average frequency of follow-up questions was 2.30 ($SD = 3.21$) in the social connectedness condition, and 1.45 ($SD = 2.21$) in the control condition. Figure 3 presents averages using the raw, non-transformed variables for interpretability. However, there was no statistically significant effect of the intervention on the amount of verbal validation during the interaction ($\beta = 0.07$, $b = 0.10$, 95% CI [−0.05, 0.19], $p = 0.234$; see Fig. 3). On the raw (non-transformed) scale, the average frequency of verbal validation was 7.49 ($SD = 6.36$) in the social connectedness condition, and 6.90 ($SD = 5.45$) in the control condition. The pattern of results was unchanged in unadjusted models (see Supplementary Table S1.4).

**Exploratory mediation model**: We next explored whether participants' increased follow-up questions mediated the effect of the intervention on a latent factor that indexed the quality of social connection (Exploratory Hypothesis). An initial measurement model of all social connection indicators (i.e., participant and partner conversational response times, BIPR and participant- and partner-reported perceived positivity resonance) demonstrated poor fit ($\chi^2(5) = 33.471$, $p < 0.001$, CFI = 0.834, RMSEA = 0.141, SRMR = 0.076). Modification indices suggested that allowing participant and partner conversational response time to covary would improve model fit. The adjusted model fit significantly better than the original ($\Delta\chi^2(1) = 25.73$, $p < 0.001$), and showed overall good fit ($\chi^2(4) = 7.738$, $p = 0.102$, CFI = 0.978, RMSEA = 0.057, SRMR = 0.029). All social connection indicators significantly loaded on a single latent factor ($ps < 0.01$; see Supplementary Fig. S1.1 for full CFA results). However, because some bootstrap runs failed to converge when integrating the latent factor into the mediation model, we opted for a simpler model based on extracted factor scores. This allowed us to retain the estimated latent variable while achieving a stable and well-fitting mediation model. Supporting our exploratory hypothesis, we found that being in the social connectedness intervention (versus the active control) predicted the quality of social connection via increased follow-up questions (indirect effect: $\beta = 0.04$, $b = 0.06$, 95% CI [0.012, 0.124], $p = 0.031$; direct effect: $\beta = 0.03$, $b = 0.04$, 95% CI [−0.135, 0.214], $p = 0.630$; see Fig. 4). In support of our hypothesized temporal order, no significant indirect effect emerged in the alternative model in which social connection served as the mediator between intervention condition and observed follow-up questions (indirect effect: $\beta = 0.02$, $b = 0.03$, 95% CI [−0.022, 0.081], $p = 0.243$; direct effect: $\beta = 0.13$, $b = 0.19$, 95% CI [0.028, 0.353], $p = 0.025$).

Overall, Study 1 provides evidence that high-quality listening behaviors, as indexed by two distinct behaviorally coded verbal indicators during semi-structured "deep talk" with a stranger, are associated with markers of high-quality social connection, and are increased by a social connectedness intervention (follow-up questions only). Furthermore, evidence emerged

that increased follow-up questions following the social connectedness intervention mediated improved social connection quality, assessed as a latent factor. However, Study 1 was limited by testing hypotheses in a highly controlled environment, using a task designed to elicit closeness. Additionally, we focused only on verbal listening behaviors, which are thought to provide the strongest signal, rather than the broader set of observable behaviors associated with high-quality listening. In Study 2, we aimed to test a similar set of hypotheses in a small talk context while also incorporating a global behavioral measure of high-quality listening that holistically encompasses both verbal and nonverbal components.

## Study 2
Descriptives for study variables are presented in Table 3.

**Validation hypothesis.** In support of our Validation Hypothesis, we found that the global listening behavioral coding scheme that encompasses verbal and nonverbal components was positively correlated with the log-transformed frequency of follow-up questions ($r (344) = 0.54$, 95% CI [0.47, 0.61], $p < 0.001$) and verbal validation ($r (344) = 0.67$, 95% CI [0.61, 0.73], $p < 0.001$). Verbal validation and follow-up questions were also positively correlated ($r (345) = 0.32$, 95% CI [0.23, 0.41], $p < 0.001$).

**Hypothesis 1: Associations between listening behaviors and markers of social connection. Conversational response time:** Supporting our hypothesis, participants coded as displaying greater (vs. fewer) global listening behaviors also had significantly faster conversational response time and interacted with a partner with faster response time (participant: $\beta = -0.14$, $b = -0.04$, 95% CI [−0.08, −0.01], $p = 0.014$, see Fig. 5A; partner: $\beta = -0.16$, $b = -0.05$, 95% CI [−0.09, −0.02], $p = 0.003$, see Fig. 5A). Replicating Study 1, asking more follow-up questions was significantly associated with interacting with partners with faster

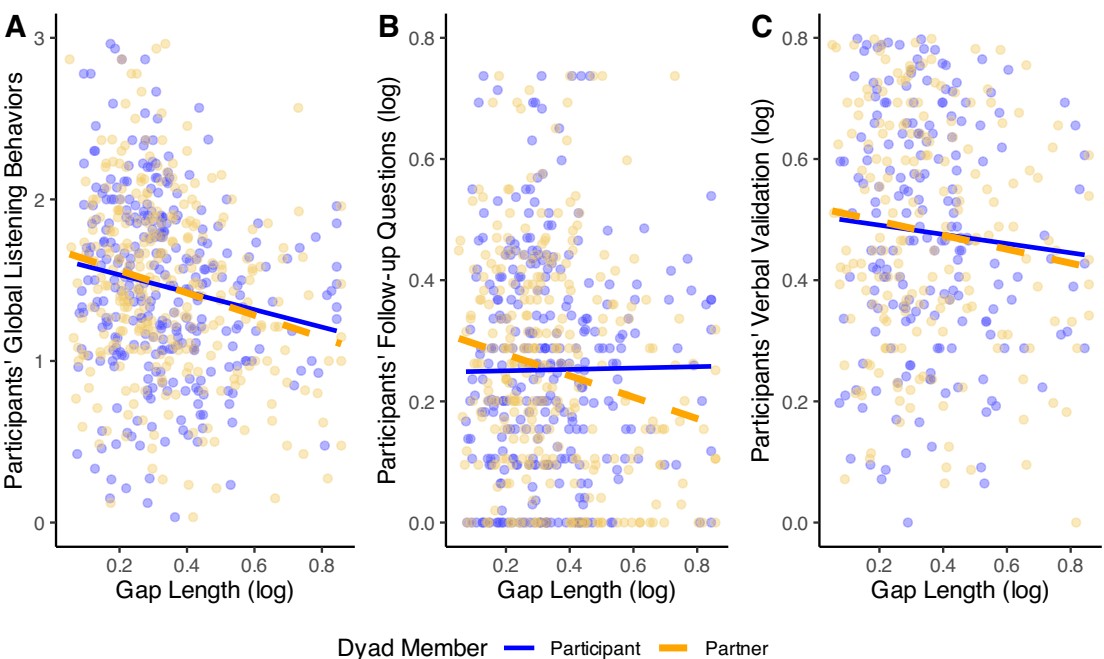

**Fig. 5 | Study 2: Participants' listening behaviors and conversational response time.** Figures display results of regression models testing global listening behaviors (**A**, left, $N = 302$), follow-up questions (**B**, middle, $N = 303$) and verbal validation (**C**, right, $N = 303$) on participant and partner response times.

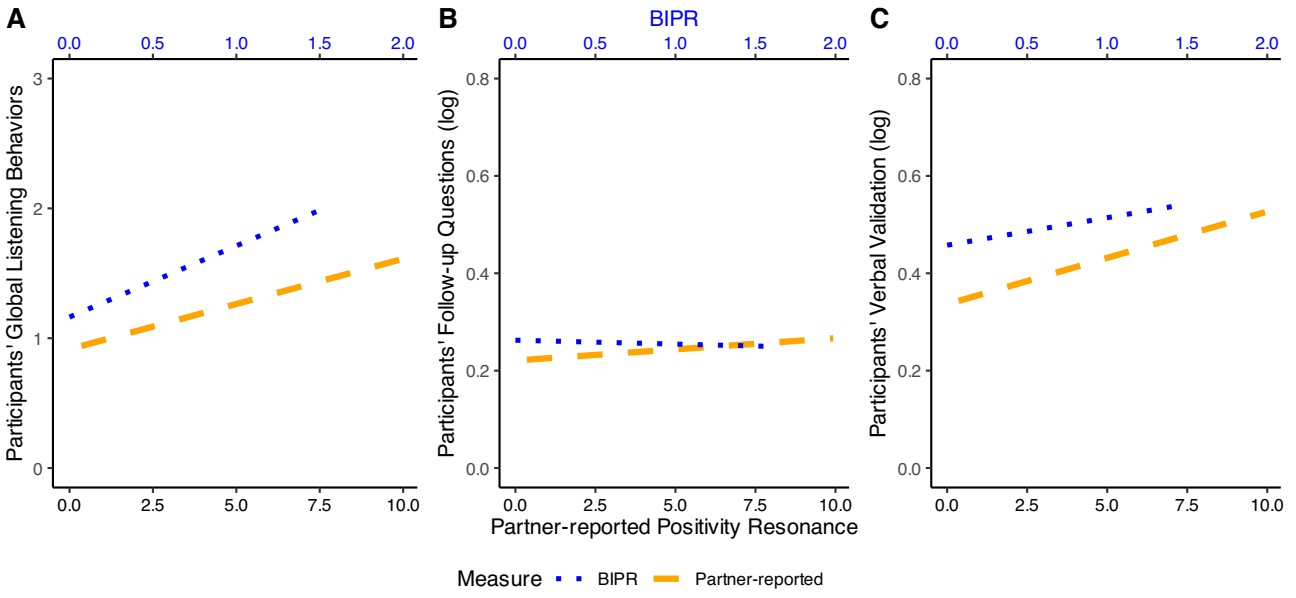

**Fig. 6 | Study 2: Participants' listening behaviors and positivity resonance.** Figures display results of regression models testing participants' global listening behaviors (**A**, left, $N = 347$), follow-up questions (**B**, middle, $N = 348$) and verbal validation (**C**, right, $N = 348$) on BIPR and partner-reported perceived positivity resonance. Scatter points were omitted to avoid visual confusion arising from the dual $x$-axes (Partner-reported and BIPR measures), which are scaled differently.

conversational response time (partner: $\beta = -0.18$, $b = -0.18$, 95% CI [$-0.29$, $-0.07$], $p = 0.001$; see Fig. 5B). Unlike Study 1, asking more follow-up questions was not significantly associated with participant's own response time (participant: $\beta = -0.01$, $b = -0.01$, 95% CI [$-0.11$, 0.10], $p = 0.886$; see Fig. 5B). Additionally, participants who displayed greater verbal validation behaviors also had significantly faster conversational response time ($\beta = -0.22$, $b = -0.12$, 95% CI [$-0.17$, $-0.06$], $p < 0.001$; see Fig. 5C), whereas their effect on partner response time was not statistically significant ($\beta = -0.10$, $b = -0.06$, 95% CI [$-0.12$, 0.01], $p = 0.079$; see Fig. 5C). Patterns of results were unchanged in unadjusted models (see Supplementary Table S2.1).

**Positivity resonance**: Supporting our hypothesis, participants who displayed more global listening behaviors also had significantly greater behaviorally coded (BIPR) and partner-reported positivity resonance (BIPR: $\beta = 0.34$, $b = 0.20$, 95% CI [0.14, 0.26], $p < 0.001$; Partner-reported positivity resonance: $\beta = 0.21$, $b = 0.64$, 95% CI [0.34, 0.95], $p < 0.001$; Fig. 6A). Unlike the deep conversation context of Study 1, during small talk, the number of follow-up questions asked by the participant was not significantly associated with either behaviorally coded or partner-reported positivity resonance (BIPR: $\beta = -0.02$, $b = -0.03$, 95% CI [$-0.22$, 0.16], $p = 0.763$; Partner: $\beta = 0.06$, $b = 0.54$, 95% CI [$-0.47$, 1.56], $p = 0.295$; Fig. 6B). Replicating Study 1, participants who engaged in greater verbal

validation did show significantly greater behaviorally coded and partner-reported positivity resonance during the small-talk interaction (BIPR: $\beta = 0.20$, $b = 0.21$, 95% CI [0.10, 0.32], $p < 0.001$; Partner: $\beta = 0.26$, $b = 1.41$, 95% CI [0.85, 1.98], $p < 0.001$; Fig. 6C). Patterns of results were unchanged in unadjusted models (see Supplementary Table S2.2).

**Hypothesis 2: Effect of social connectedness interventions on listening behaviors.** **Social connectedness interventions**: Next, we tested whether participants randomized to a behavioral intervention group, compared to the passive control, showed increased listening behaviors during a small-talk interaction with a stranger. Across all three measures of listening, no statistically significant effects emerged (global: $\beta = 0.00$, $b = 0.00$, 95% CI [−0.13, 0.13], $p = 0.957$; follow-up questions: $\beta = 0.08$, $b = 0.03$, 95% CI [−0.01, 0.07], $p = 0.152$; verbal validation: $\beta = 0.05$, $b = 0.04$, 95% CI [−0.04, 0.11], $p = 0.350$). For descriptives of listening variables by condition, see Supplementary Table S2.3. The Supplementary Table S2.4 presents individual effects of all four conditions (i.e., not grouped into one behavioral intervention group) compared to the passive control. Given that we did not find statistically significant support for intervention effects in this context, we did not proceed to test our exploratory mediation model.

## Discussion

We undertook the current work to test whether objectively assessed high-quality listening behaviors during conversations between strangers predict markers of high-quality social connection. Whether in semi-structured "deep talk" (Study 1) or during a "small talk" conversation that better approximates real-world interactions with strangers (Study 2), we found consistent support for our preregistered Hypothesis 1, that higher-quality listening behaviors would be associated with higher-quality social connection. In Study 1, verbal validation and follow-up questions were linked to both partners' conversational response times and to behaviorally coded and partner-reported positivity resonance, but we did not find evidence linking listening behaviors to participant's self-reported positivity resonance. In Study 2, we replicated this pattern and found that global listening behaviors (i.e., a holistic synthesis of verbal and nonverbal indicators) were associated with all tested markers of social connection (i.e., both partners' conversational response time and behaviorally coded and partner-reported positivity resonance; participant-reported positivity resonance was not measured in Study 2). However, in the small talk context of Study 2, we did not find evidence that verbal validation was linked to *partner* conversational response times. Furthermore, unlike Study 1, participants' follow-up questions were only significantly associated with partner response time (no evidence for the questioner's own response time or positivity resonance).

Corresponding with past evidence that one's own feelings of connection may be better predicted by their partner's behaviors (e.g., partner response time[22]), our pattern of findings suggested that high-quality listening does not always translate to markers of social connection for listeners themselves. These findings may also reflect that these two markers of connection quality capture distinct aspects of connection, with the speed of one's conversational responses representing an individual-level marker and positivity resonance representing an emergent collective marker (here, at the dyad-level). In a bid to connect, participants displaying high-quality listening might also respond more quickly to their partner, yet in instances when a partner does not reciprocate high-quality listening, participants may not come to experience a collective state of connection (i.e., togetherness in ELT or positivity resonance). This underscores the dyadic and reciprocal nature of listening in conversations: the collective state of connection between members of a dyad may be most pronounced when both individuals mutually display high-quality listening behaviors to each other.

In the brief small-talk context of Study 2, the global listening behavior emerged as the more consistent predictor of social connection compared to verbal indicators. Although verbal indicators may provide the strongest evidence of listening to the speaker[12,13], high-quality listening is a construct made up of a combination of behaviors, among which no single behavior

sufficiently indicates high-quality listening[9]. Accordingly, explicit verbal indicators may be most indicative of listening when accompanied by a range of nonverbal cues (e.g., nodding, eye contact), particularly in settings when there is greater ambiguity regarding the degree to which unaffiliated partners may be connecting (i.e., small talk vs. deep talk).

In Study 1, we found evidence that an intervention for which participants were instructed to connect with strangers and acquaintances over 48 h, subsequently caused them to ask a greater number of follow-up questions during semi-structured conversation, which in turn, predicted increased social connection assessed as a latent factor. Although this might suggest that the intervention's success in increasing connection quality may be partly explained by participant's asking more follow-up questions, we hasten to underscore that effect sizes were small, suggesting that additional processes beyond listening behaviors also contribute to the intervention's impact on improved social connection (e.g., personality, relational dynamics, motivation). Notably, examination of the conversation transcripts revealed that individuals scored as high-quality in their listening often recalled specific details shared by their partners and referenced earlier content to progress the conversation with follow-up questions (e.g., "What were you thinking when the professor made that comment?", "What happened after you told her what was on your mind?"). Given that the intervention did not provide explicit instruction about listening or asking questions, our findings suggest that people may intuitively attend more closely to speakers and ask more questions to more fully understand their partner as a route to connection. Furthermore, the finding that follow-up questions predicted increased feelings of connection suggests that mutual self-disclosure alone may be insufficient for creating interpersonal closeness. Finally, we did not find evidence for the reverse indirect effect, i.e., a latent factor of social connection did not explain increased follow-up questions. This is consistent with claims from ELT[26] and Positivity Resonance Theory[23] that high-quality listening and its theorized correlates[8] (i.e., perceived safety and real-time sensory connection) are conducive conditions for high-quality connection. Future work could strengthen evidence for the indirect effect of the social connectedness intervention on indicators of social connection by experimentally manipulating follow-up questions, thereby establishing temporal precedence and causality.

However, in the context of "small talk" (i.e., Study 2) following a 35-day connectedness intervention, we did not find support for our hypothesis that a social connectedness intervention would increase listening behaviors. Intervention effects may have differed across the two studies for multiple reasons. A brief small talk interaction with an experimenter is likely a significantly higher bar to observe changes in behavior compared to a lengthier mutual-disclosure task with a peer (as in Study 1). It may be that follow-up questions in the context of "small talk" were sometimes offered out of politeness or as a way to combat potentially sitting in awkward silence. Conversely, in Study 1, the mutual-disclosure task minimizes a common barrier to connecting with a stranger by providing clear topics of discussion, minimizing conversational lulls. Thus, follow-up questions may have been less likely to be motivated by awkward silence and more likely to reflect motivation to connect. Another possibility is that the intervention in Study 1 had a more effective and immediate influence on social behavior than the interventions in Study 2. For one, unlike Study 1, there is no reported evidence that the intervention in Study 2 predicted in-lab social connection[28]. Whereas the intervention in Study 2 consisted of 35 days of brief, unchanging daily emailed reminders to connect, Study 1 was a short, 48-h intervention with several advanced design features to enhance effectiveness. These advanced design features included having participants interact with a virtual human to discuss opportunities and obstacles for connection and to create behavioral implementation intentions, each known to facilitate behavior change[30,38,39]. Thus, the design in Study 1 may have led to a more robust test of immediate changes in social behavior.

Consistent with ELT, we anticipate findings regarding high-quality listening behaviors and positivity resonance to generalize across social contexts and relationships, or as previously shown with perceived listening, between strangers discussing disagreement[26]. It would be interesting to test

the extent to which the strength of the association between high-quality listening behaviors and positivity resonance may vary depending on the closeness of interacting partners. In established relationships, judgments of connection can be partly informed by past experiences. In contrast, strangers must rely solely on observable behaviors during the interaction to form their impressions, making these behaviors potentially more influential. One might also argue that explicit verbal expressions of high-quality listening may come more naturally between strangers. For instance, experimental evidence finds that people tend to communicate the meaning of ambiguous phrases more effectively to a stranger, compared to a close other[40], likely because they may actively monitor and consider the stranger's perspective, with greater recognition that it may diverge from their own. Therefore, people might present more explicit signals of listening to strangers, either to clarify their own positive intentions or signal openness to the other's perspective. Furthermore, as longstanding research suggests that gaining information about novel stimuli is intrinsically rewarding[41], curiosity and interest are likely naturally heightened during conversation with a novel partner[42], potentially leading to more opportunity and motivation to ask follow-up questions.

It is unclear the extent to which findings regarding conversational response time would similarly generalize to other social contexts, such as within close relationships or during disagreements between strangers. On one hand, one might expect that in potentially hostile contexts, faster responses may predict lower-quality listening. On the other hand, recent research shows that people tend to overestimate how negative a disagreement discussion with a stranger may be, as well as underestimate the extent to which they would find common ground[43]. This suggests that dyads who display more verbal indicators of high-quality listening may also respond faster to one another, compared to dyads with fewer verbal indicators. It is also possible that the effect of high-quality listening behaviors on faster conversational response time may be more consistently observed in conversations between strangers (vs. close others), as *slower* response times between close others have also been found to signal heightened connection in certain contexts, whereas strangers are more likely to report gaps in conversation as awkward[32]. Future work is needed to understand how listening behaviors and response time dynamics may unfold dynamically during conversations across social contexts.

## Limitations

One important limitation is that the "stranger" in both studies was either a trained confederate (Study 1) or an experimenter (Study 2). Although this allowed for greater experimental and logistical control, these partners likely became increasingly experienced in their respective roles, potentially diminishing the authenticity of engaging with a novel partner. Nonetheless, interactants were strangers, and the interaction may have been experienced as novel to the participant despite their partner's growing familiarity with the role. We attempted to minimize the potential influence of individual differences across confederates/experimenters by statistically controlling for confederate/experimenter effects in our primary models. However, the present study warrants replication and extension in naturalistic contexts with unfamiliar partners who are not trained or practiced in the study protocol. It is important to note that confederates and experimenters were blinded to participants' randomized condition and, given the secondary use of these datasets, were unaware of possible study aims related to conversational response times or listening behaviors. An additional limitation related to the use of confederates and experimenters is that their self-reported positivity resonance may also be influenced by their subjective comparisons of previous participant interactions. However, this concern is mitigated by the fact that we also used objective (conversational response times) and behavioral (BIPR) measures to converge on our construct of interest, and therefore, do not rely on partner reports as the sole basis for our claims. Finally, another limitation of the study was that both samples were largely female and recruited from the same university setting. Although samples had a diverse age range, our sample is not fully representative of the larger U.S. or world population.

## Conclusion

Social connections with strangers are a unique interpersonal context that is known to independently predict mental health and belongingness[2]. Yet, for many, connecting with a stranger can initially feel challenging or anxiety-provoking[6]. Consistent with theory, our research finds evidence that observable listening behaviors are associated with multiple markers of social connection, including the speed at which people respond to one another in conversation and people's perceptions of the shared emotional quality of the connection. We additionally find that one practical route to increasing social connection during an interaction is asking relevant follow-up questions. Evidence that listening behaviors accounted for (statistically mediated) the beneficial effects of a social connectedness intervention illuminates a promising direction for optimizing future behavioral interventions. In sum, high-quality listening behaviors may be foundational to fostering high-quality connection between strangers.

## Data availability

We report how we determined our sample sizes and all data exclusions and manipulations used in both studies. All data used for analyses are available on OSF: https://osf.io/2xbwt/?view_only=25e76a885eee4b8db51342b893ac0f78.

## Code availability

All code are publicly available at https://osf.io/2xbwt/?view_only=25e76a885eee4b8db51342b893ac0f78.

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

## Acknowledgements

We thank the dozens of PEP lab research assistants (2019–2024) who served as experimenters, confederates or behavioral coders for the current studies, and thus made this research possible. This research was also supported by grants from the Templeton World Charity Foundation (Study 1: TWCF0472; Study 2: TWCF0325) awarded to Barbara L. Fredrickson. The funders had no role in study design, data collection and analysis, decision to publish, or preparation of the manuscript.

## Author contributions

Taylor West developed the study concept, developed coding methods, collected and analyzed the data, and drafted and revised the manuscript. Sara Huston contributed to the study concept, developed coding methods and led the project administration, including coding data collection. Kylie Chandler contributed to the study concept and coding method development. Jieni Zhou contributed to coding method development and coding data collection. Barbara Fredrickson supervised the project and revised the manuscript.

## Competing interests

The authors declare no competing interests.
