## [Transparent Peer Review file · Communications Psychology]

High-Quality Listening Behaviors Linked to Social Connection Between Strangers

Corresponding Author: Dr Taylor West

Version 0:

Decision Letter:

Dear Dr West,

Thank you for your patience during the peer-review process. Your manuscript titled "High-Quality Listening Behaviors Linked to Social Connection Between Strangers: Implications for Optimizing Behavioral Interventions" has now been seen by 2 reviewers, and I include their comments at the end of this message. They find your work of interest but raised some important points. We are interested in the possibility of publishing your study in Communications Psychology, but would like to consider your responses to these concerns and assess a revised manuscript before we make a final decision on publication.

We therefore invite you to revise and resubmit your manuscript, along with a point-by-point response to the reviewers. Please highlight all changes in the manuscript text file.

Editorially, we ask that you carefully address the reviewers' concerns regarding methodological clarifications and inconsistencies in the argument. Please also provide more information about the use of confederates, as raised by both reviewers.

Please ensure you follow our statistical guidelines when reporting statistics (<https://www.nature.com/commpsychol/submit/submission-guidelines#statistical-guidelines>). Please note in particular our requirements for the reporting and interpretation of null-results. Non-significant findings derived from null-hypotheses significance tests should be reported in full, but may not be interpreted. Where you interpret null results, this interpretation must be based on Bayes Factors or equivalence tests.

Please add a power sensitivity analysis to your manuscript. We ask that you do not conduct a post-hoc power analysis based on the observed effect size in your study (cf. Lakens, 2022, <https://doi.org/10.1525/collabra.33267>).

Last, please note that we allow unlimited space for Methods reporting.

I am attaching an Editorial Requests Table that details critical reporting requirements for the revised manuscript. Please attend to each item and ensure your manuscript is fully compliant. If your revised manuscript is not aligned with these requests on major issues, such as those concerning statistics, it may be returned to you for further revisions without re-review.

Please submit the following items:

- Revised manuscript
- Point-by-point response to the referees' comments

- Cover letter (as a separate document)
- <https://www.nature.com/documents/nr-reporting-summary.zip>>Nature Research Reporting Summary
- <https://www.nature.com/documents/nr-editorial-policy-checklist.pdf>>Editorial Policy Checklist
- Completed Editorial Request Table (attached).

via this link: Link Redacted .

Additional guidance is available in our style and formatting guide <https://www.nature.com/documents/commpsychol-style-formatting-guide-accept.pdf>>Communications Psychology formatting guide.

Best regards,

Yafeng Pan

Yafeng Pan, PhD
Editorial Board Member
Communications Psychology
orcid.org/0000-0002-5633-8313

REVIEWER EXPERTISE:

Reviewer #1: social connection, behavioral coding

Reviewer #2: social connection, behavioral coding

REVIEWER REPORTS:

Reviewer #1 (Remarks to the Author):

West et al. report evidence that specific markers of "high-quality" listening have the potential to increase social connection among strangers. Although claims abound in the academic literature and popular press, the authors rightly admonish the reader for taking those claims at face value. At the same time, I would like to propose that there are a few published studies that might be valuable to include as they point to "specific, actionable behaviors that individuals can adopt to effectively signal high-quality listening to their interaction partners." Indeed, before Collins (2022) "argued" that "verbal cues ... [are] the strongest signal that conveys high-quality listening to a speaker," research conducted in my lab while I was at Louisiana State University found that specific verbal markers of "supportive listening" were "more important to the prediction of supportive conversation outcomes than their nonverbal counterparts" (Bodie et al., 2015, p. 166). Granted, these conversations involved troubles-talk and not small talk, but they were among strangers (some of whom were trained in active listening, but not all). We cite some other studies in that article that might also provide useful support for the claims the authors make in their justification for why their studies are needed. To be sure, my past work does not answer all of the questions this article asks; nevertheless, it is also not completely irrelevant. The authors might also find useful the coding procedures we have used in our past work. There are standard measures and rubrics for various listening behaviors, raising the question why the authors were compelled to create one anew for these studies.

One methodological question I have about the manuscript is about the use of confederates, not because I am against their use but because the claims of this manuscript are about interactions between strangers. To be sure, the confederates in this

study were strangers to the participants; and, yet, it is unclear how that may or may not matter for the primary claims the manuscript seeks to make (namely that listening behaviors contribute to markers of social connection). Perhaps the concern is less relevant for the more "objective" markers like response time compared to relying on confederate reports of social connection to inform our understanding of what role listening plays in its establishment. What does it mean to have a person affiliated with a study submit data that informs the inferences we make about that study? It seems there may be a conflict of interest there. In any case, I don't view this as a fatal flaw or a reason to reject the manuscript, but it does seem to warrant more than a passing remark in the section on limitations.

I would like to see more information about the confederates used in the study - how many were there, and what does it mean to "control" for their effects? Was there a dummy-coded variable included in the analysis? Maybe I missed something, but I was a bit confused on these details.

I'm also not entirely clear on the magnitude of the effects found for listening behaviors. There are no means reported, except in figures where I have to assume values rather than know explicitly. Of course, understanding differences in values for some variables is complicated by the fact that they were transformed. But, for instance, in Figure 3, it seems like compared to the active control, those in the social connectedness condition(s) asked, on average, between 1 and 1.5 more questions. Understanding that there was a statistically significant difference is one thing; understanding whether the difference is meaningful is another. Why does it matter that I ask 1 (or 1.5) more question over a 10 minute conversation? Does that really stand to make a difference?

Lastly, and related to both verbal markers of listening (questions and validation), there is a difference between "asking a question" and the kind of question (or "validating emotion" and doing so explicitly vs. implicitly). Constructs like verbal person centeredness can help unpack what it means to do validation more/less well, and there are similar models of what constitute "good" questions. Perhaps the authors might provide some insight in the Discussion around how listeners might engage in some of these behaviors by looking at the transcripts and inferring the kinds of rhetorical strategies that could be potentially useful to train (so that we create particularly "high" levels of "high-quality" listening).

Reviewer #2 (Remarks to the Author):

Thank you for the opportunity to review the manuscript "High-Quality Listening Behaviors Linked to Social Connection Between Strangers: Implications for Optimizing Behavioral Interventions". While this manuscript touches on an important topic and has interesting findings, I see some major issues that need to be addressed before I recommend publication. I therefore recommend major revisions.

Abstract:

- The authors state that they test theory-driven behavioral and subjective markers of social connection, but from the main text I understood that these were also at least partially empirically proven in previous studies.
- The social connectedness interventions can be introduced more.
- It is currently not clear that "asked their partner more follow-up questions" effectively means shows more listening behaviors.

Introduction:

- Nice opening to emphasize the importance of interactions with strangers.
- Line 73: The authors state that "people need simple, actionable strategies to improve connection in their daily interactions", but it is not clear how you come to this conclusion based on the previous paragraph, and references are missing.
- Line 96 – 102: Backchannel signals are also verbal, so what is meant by the explicit verbal expressions? Maybe an example could help here.
- Line 102 - 105: "Between strangers, explicit verbal indicators may be especially useful for building connection, by helping interactants better anticipate one another's thoughts and intentions, reducing uncertainty, signaling positive intention, and establishing interpersonal trust when there were no prior opportunities to do so." This sentence needs more explanation and substantiation with references: I did not previously read about the uncertainty reduction and trust fostering nature of listening and why that is the case.
- Line 179: The social connectedness interventions, later connection-based interventions, appear out of the blue here. What are they? This needs to be explained.
- I feel there are some inconsistencies in the argumentation: first the authors mention that there has been a lot of listening research looking at indicators of social connection, but that this is mostly based on experimental data where listening is manipulated, so less naturalistic, but later (line 184) you say that most research is based on self-report perceptions of listening, which sounds rather naturalistic.

Study 1:

I have a couple of questions concerning the methods of Study 1 (and 2) that need to be clarified in the manuscript:

- I understand that "Sample size was determined based on a power analysis for the original aims of the grant-supported study". But I would still like to know what your power was for the current study. This also applies to Study 2.
- Did students sign informed consent for Study 1 and 2? This should be explicitly stated in the manuscript. Also, was there a cover story for participants and what was this?
- Why was the diaphragmatic breathing intervention chosen as active control group?
- What was the short educational video (watched before interacting with Ellie) about?
- Did students keep to the assignment they were given to practice in the intervening 48 hours? I think this is an important aspect of both Study 1 and 2 (with its 35 days interval) since that is the social connectedness intervention the authors aim to

test.

-Why did expressing simple agreement (e.g., "me too") not count as verbal validation? (line 278-279).

-Line 285: How did the authors decide when reliability was met in the coder training?

-Line 286: The authors state that to assess reliability, all coders coded 20% of the videos, but with 5 coders, this comes down to 100% of the videos, so 0 overlap right? I think this is a phrasing issue, since the authors probably mean to say that a set of 20% of the videos were coded by all 5 coders.

-Line 298: The authors state that "There were 16 "unusable transcripts with major errors", primarily due to poor audio quality. It is unclear what the authors did with these transcripts. Were they excluded from the analysis, and also from the listening coding that was described earlier? I ask the authors make this clear and, if it applies to all qualitative analysis, mention it earlier.

-I think it is not entirely fair to treat the confederate's and later experimenter's (as the interaction partner) experiences and behaviors as 'naturalistic', since both of them did know the purpose of the study, I assume. Moreover, and more importantly, they were partly instructed to behave in a certain way and I would assume that they developed some behavioral patterns during the many interactions they had. The authors do reflect on this in the Discussion section, which I appreciate, but it is not entirely clear how they "controlled for confederate/experimenter effects" in their primary models. Can they elaborate on that? I would also like a mention of this limitation earlier on, when writing about the study designs.

-I would like to see some descriptives, either in the Results or in the Methods of the average positivity resonance as well as the behavioral coding, to get a general impression of the dataset. For the rest, as far as I can judge, the analyses appear sound and the results clear.

Study 2:

-See comments under Study 1.

-I ask the authors to indicate the number of participants per treatment (and control) group. And to reflect on the fact that the groups they later compare (all treatment groups vs. the one control group) probably are very different in size.

-I am not sure but isn't it rather unusual to refer to another paper (about the same dataset) for more information about the study setup? I would assume the current manuscript should be a stand-alone paper and I would therefore recommend the authors to include an appendix or online materials with these same details.

-I would also recommend rephrasing "naturalistic context" to something more fitting, since, as I indicated before, I find this setting with an experimenter as interaction partner not very naturalistic. Otherwise, the authors should clearly explain how this context is naturalistic.

-I would like to know why only the experimenter reported on positivity resonance and the participants did not (since this is an important limitation).

-I am not sure whether I understand why the non-verbal listening cues were now included after all, since the authors clearly defended their focus on verbal listening in the Introduction. Can the authors clarify this?

Discussion:

-Some of the findings described are not clear for readers that skip or skim the Results section. I recommend to clarify this by repeating the main findings instead of simply stating that they were "replicated" as is done now (lines 712-714).

-I miss speculation for why the authors find an effect for the intervention in Study 1 and not in Study 2 (maybe the different time duration can be a factor?).

Minor points:

-Prevent abbreviations, in line with APA guidelines, so write out things like "needn't" (line 72).

-I recommend a thorough read through because there are some writing mistakes and missing words, e.g., line 198: "listening behaviors are raised by an intervention designed improve social connection". In this sentence the word "to" is missing, also I am wondering whether "raised" which is used more in this context is the right term.

Version 1:

Decision Letter:

Dear Dr West,

Thank you for your patience during the peer-review process. Your manuscript titled "High-Quality Listening Behaviors Linked to Social Connection Between Strangers: Implications for Optimizing Behavioral Interventions" has now been seen by 1 reviewer, and I include their comments at the end of this message. They find your work of interest but raised some important points. We are very interested in the possibility of publishing your study in Communications Psychology, but would like to consider your responses to these concerns and assess a revised manuscript before we make a final decision on publication.

We therefore invite you to revise and resubmit your manuscript, along with a point-by-point response to the reviewers. Please highlight all changes in the manuscript text file.

Editorially, we consider you address the remaining important issues on methodology (e.g., reporting the exact reliability estimates) and the results (e.g., some small and inconsistent effects between the two studies), as suggested by the reviewer. Please also streamline your Discussion.

I am attaching an Editorial Requests Table that details critical reporting requirements for the revised manuscript. Please attend to each item and ensure your manuscript is fully compliant. If your revised manuscript is not aligned with these requests on major issues, such as those concerning statistics, it may be returned to you for further revisions without re-review.

Please submit the following items:

- Revised manuscript
- Point-by-point response to the referees' comments
- Cover letter (as a separate document)
- <https://www.nature.com/documents/nr-reporting-summary.pdf> Nature Research Reporting Summary
- Completed Editorial Request Table (attached).

via this link: Link Redacted .

Additional guidance is available in our style and formatting guide <https://www.nature.com/documents/commpsychol-style-formatting-guide-accept.pdf> Communications Psychology formatting guide.

Best regards,

Yafeng Pan

Yafeng Pan, PhD
Editorial Board Member
Communications Psychology
orcid.org/0000-0002-5633-8313

REVIEWER EXPERTISE:

Reviewer #1: social connection, behavioral coding

REVIEWER REPORTS:

Reviewer #1 (Remarks to the Author):

I've completed my review of "High-quality listening behaviors linked to social connection between strangers." Overall, I believe the manuscript stands to inform us about "simple, actionable strategies to improve connection in their daily interactions" (p. 3). As a listening scholar, I appreciate the attention this article might bring to how we conceptualize, measure, and teach such an important (yet often underappreciated) life skill.

Although in our age of electronic articles "journal space" is typically not an issue, I would still encourage the authors to find places to streamline arguments and delete text. For instance, information on page 8 ("In the present study...") is largely repeated on page 12 ("...we drew from two large, archival datasets ..."). My recommendation is to reduce the 9-paragraph introduction to 4-5 paragraphs. Within that introduction, when the authors speculate about reasons that "explicit verbal expressions" operate as "stronger" indicators of "high-quality listening" they might benefit from the logic used by Herb Clark in his 1996 book, *Using Language*. In that book, Clark discusses "evidence of understand" in his model of discourse understanding (and how "grounding" happens). In that discussion, he outlines various types of evidence that listeners use to signal they have understood the speaker "well enough for current purposes." When listeners use "explicit verbal expressions" they are more directly signaling their understanding (versus when they use backchannels or other short utterances). So, while it is true that "backchannel responses ... can be deceptively used when people merely pretend to listen" that is not the only explanation (and in my opinion not the best explanation) for why these cues are weaker signals that one is listening.

In both studies, I recommend the authors report the exact reliability estimates - for instance in the second footnote on page 10 (instead of $ICC > .80$, report the actual reliability estimate obtained) and on page 14 (once reliability was met, $ICC = ??$). In that same paragraph on page 14, I am not clear as to whether the "same set of videos" was coded "each week" (that is whether each coder coded videos 1, 2, and 3 or whether one coder coded 1-3 and the other coded 4-6). I also was confused by how an "average frequency of follow-up questions and verbal validation across coders were used for analysis" if "only 20% of the videos were coded by all 5 coders." How is there an average if 80% of the videos were coded by a single coder?

I am particularly interested in the data presented as Figure 3. There appears a great deal of variability in the number of questions (as well as the amount of verbal validation), and yet experimental condition only explained a small percent of that variance (helpful for me to look at d values which were all less than .5, corresponding to an r -squared of less than .05). What explains the remaining variability - it does not seem that this study tracked the extent to which people participated in the intervention; but perhaps practicing everyday vs. not much practice may have something to do with that variability? On page 26 (Study 2), the authors mention "daily reporting" - are there data that could answer this?

Those distributions are also interesting insofar as the mean # questions was quite low compared to verbal validation. Is this a function of presence/absence of these variables (conceptual question) or how they are operationalized (measurement question)? There seem to be more ways to express verbal validation but only one way to ask a question - and as the literature on verbal response modes (largely in the context of therapy) suggests, the nature of a question can be grammatical or pragmatic/use-based (meaning I can "ask a question" even if my wording is not phrased grammatically as a question; e.g., I could say, You're leaving already with rising intonation for instance). In any event, the fact there is so much variability in behaviors left to be explained seems in need of an explanation. Overall, these are TINY effects (when they occur), and the effects that are there are not fully consistent (especially between the two studies). It is, thus, not 100% accurate to claim, for instance, that "people may intuitively ask their partner more follow-up questions to increase connection" - verbal validation was much more common (hence more intuitive); plus the increase was very very very small.

I am also a bit concerned that the inclusion of "both verbal and non-verbal components" of listening in Study 2 was largely driven by the fact that the coding of non-verbal behaviors in Study 1 failed. There are several places between the end of Study 1 and the beginning of Study 2 that the authors present this as an advancement of Study 2, but in reality it was something they thought of already for Study 1 (but it just did not work out).

Like the introduction, I also think the Discussion is a bit overwritten. The authors might also usefully combine many aspects of Study 1 and Study 2 to try and reduce the length of the manuscript. They both answer a similar set of questions, and the methods are quite similar.

Graham Bodie

Version 2:

Decision Letter:

Dear Dr West,

Your manuscript titled "High-Quality Listening Behaviors Linked to Social Connection Between Strangers" has now been editorially reviewed, and I am delighted to say that we are happy, in principle, to publish a suitably revised version in Communications Psychology.

We therefore invite you to revise your paper one last time to address the remaining editorial requests. At the same time we ask that you edit your manuscript to comply with our format requirements and to maximise the accessibility and therefore the impact of your work.

EDITORIAL REQUESTS:

SUBMISSION INFORMATION:

OPEN ACCESS:

At acceptance, you will be provided with instructions for completing the open access licence agreement on behalf of all

authors. This grants us the necessary permissions to publish your paper. Additionally, you will be asked to declare that all required third party permissions have been obtained, and to provide billing information in order to pay the article-processing charge (APC).

* **DATA AVAILABILITY:**

Link Redacted

Best regards,

Jennifer Bellingtier

Jennifer Bellingtier, PhD
Senior Editor
Communications Psychology

Yafeng Pan, PhD
Editorial Board Member
Communications Psychology
orcid.org/0000-0002-5633-8313

Author Responses to Reviewers

Thank you for your careful consideration of our manuscript “High-Quality Listening Behaviors Linked to Social Connection Between Strangers: Implications for Optimizing Behavioral Interventions” (COMMPSYCHOL-25-0180). Below you will find our detailed responses, with the complete and verbatim text from the reviewers presented in black, alongside our responses in purple text. All page numbers mentioned refer to the revised version of the manuscript. We have also highlighted changes directly in the manuscript, as requested.

REVIEWER EXPERTISE:

Reviewer #1: social connection, behavioral coding

Reviewer #2: social connection, behavioral coding

REVIEWER REPORTS:

Reviewer #1 (Remarks to the Author):

West et al. report evidence that specific markers of "high-quality" listening have the potential to increase social connection among strangers. Although claims abound in the academic literature and popular press, the authors rightly admonish the reader for taking those claims at face value. At the same time, I would like to propose that there are a few published studies that might be valuable to include as they point to "specific, actionable behaviors that individuals can adopt to effectively signal high-quality listening to their interaction partners." Indeed, before Collins (2022) "argued" that "verbal cues ... [are] the strongest signal that conveys high-quality listening to a speaker," research conducted in my lab while I was at Louisiana State University found that specific verbal markers of "supportive listening" were "more important to the prediction of supportive conversation outcomes than their nonverbal counterparts" (Bodie et al., 2015, p. 166). Granted, these conversations involved troubles-talk and not small talk, but they were among strangers (some of whom were trained in active listening, but not all). We cite some other studies in that article that might also provide useful support for the claims the authors make in their justification for why their studies are needed. To be sure, my past work does not answer all of the questions this article asks; nevertheless, it is also not completely irrelevant. The authors might also find useful the coding procedures we have used in our past work. There are standard measures and rubrics for various listening behaviors, raising the question why the authors were compelled to create one anew for these studies.

Thank you for highlighting your prior work on verbal markers of supportive listening. We have now added a sentence pointing to evidence that verbal indicators may be stronger predictors of listening on Page 4-5: “Consistent with this, several studies have shown that verbal markers of supportive listening were stronger predictors than non-verbal markers of positive outcomes following a conversation about upsetting events with another stranger (Bodie et al., 2015; Jones & Guerrero, 2001).”

We developed a coding scheme based specifically on Kruger & Itzchakov’s definition of high-quality listening, which emphasized positive intention and regard. This dimension is not typically reflected in existing rubrics or active listening constructs. Additionally, our coding

approach for verbal indicators uses behavioral frequency counts rather than subjective measures. We acknowledge there are certainly other coding schemes that exist in the literature, our aim was to design a coding system specifically aligned with our theoretical framework and our research context.

One methodological question I have about the manuscript is about the use of confederates, not because I am against their use but because the claims of this manuscript are about interactions between strangers. To be sure, the confederates in this study were strangers to the participants; and, yet, it is unclear how that may or may not matter for the primary claims the manuscript seeks to make (namely that listening behaviors contribute to markers of social connection). Perhaps the concern is less relevant for the more "objective" markers like response time compared to relying on confederate reports of social connection to inform our understanding of what role listening plays in its establishment. What does it mean to have a person affiliated with a study submit data that informs the inferences we make about that study? It seems there may be a conflict of interest there. In any case, I don't view this as a fatal flaw or a reason to reject the manuscript, but it does seem to warrant more than a passing remark in the section on limitations.

I would like to see more information about the confederates used in the study - how many were there, and what does it mean to "control" for their effects? Was there a dummy-coded variable included in the analysis? Maybe I missed something, but I was a bit confused on these details.

Thank you for raising this point, and we agree that the use of confederates is a limitation as they become more practiced over time. However, it is important to note that confederates and experimenters were blinded to participant condition and specific hypotheses, especially regarding listening or conversational response times, as neither outcome was part of the original study aims. In Study 1, pre-registration of these specific hypotheses did not occur until approximately 6 months following the end of data collection, and nearly four years following the end of data collection in Study 2. We were able to minimize the effect of individual differences of confederates by statistically controlling for which confederate was partnered with the participant. While practice effects remain a limitation, the fact remains that participants and confederates were strangers. Moreover, because our focus was on the *participant's* listening behaviors, and not the confederate/experimenter, we can still make valid conclusions about the participant's behaviors when interacting with a stranger. Regarding your question about having a person affiliated with the study submit data that informs inferences on the study, we appreciate this is a valid concern and question to raise. However, as you note, because we use both conversational response times (objective measure) and behavioral coded measures (by 3rd party observers), we do not rely on confederate reports to make inferences. Rather, the inclusion of these reports serve to provide another layer of evidence and convergent validity for the other measures we use in support of our claims. Removing the confederate/experimenter self-reports altogether from the present work would not change the claims we make. Overall, we do not believe using confederates undermines or biases the claims made. Even so, we note that the study warrants replication and extension in more naturalistic contexts with unfamiliar partners who are not trained or practiced in a study protocol. We elaborate on this limitation in our discussion and made the following edits to our manuscript:

- In Study 1, we specified under “Procedure” that there were 10 confederates (Page 13). Under “Analysis plan”, we briefly explained how confederate effects were controlled for in analyses (Page 17): “To minimize the effect of individual differences across confederates, we additionally included confederate identity as a categorical variable in all analyses.” Because confederates had repeated observations, we also switched all models with confederate or experimenter outcomes to nested multilevel models. While statistical estimates saw minor changes, overall findings did not change from adjusting models. This edit can be found on Page 16 and 32 under analysis plan: “Because confederates had repeated interactions, we tested effects of listening predictors on partner conversational response time and partner-reported positivity resonance using multi-level models (using the package lmerTest, Kuznetsova et al., 2017) with confederate identity as a random intercept to account for non-independence of observations.”
- In Study 2, we specified under “Procedure” that there were 4 experimenters (Page 29).
- In the discussion (Page 44-45), we have now expanded on the use of confederates: “...Although this allowed for greater experimental and logistical control, these partners likely became increasingly experienced in their respective roles, potentially diminishing the authenticity of engaging with a novel partner. Nonetheless, interactants were strangers, and the interaction may have been experienced as novel to the participant despite their partner’s growing familiarity with the role. We attempted to minimize the potential influence of individual differences across confederates/experimenters by statistically controlling for confederate/experimenter effects in our primary models. However, the present study warrants replication and extension in naturalistic contexts with unfamiliar partners who are not trained or practiced in study protocol. It is important to note that confederates and experimenters were blinded to participants condition and, given the secondary use of these data sets, were unaware of possible study aims related to conversational response times or listening behaviors..... We also note that the use of self-reported positivity resonance from the confederate and experimenter may also be influenced by their subjective comparisons of previous participant interactions. However, this concern is mitigated by the fact we also use objective (conversational response times) and behavioral (BIPR) measures to converge on our construct of interest, and therefore, do not rely on partner-reports as the sole basis for our claims.”

I’m also not entirely clear on the magnitude of the effects found for listening behaviors. There are no means reported, except in figures where I have to assume values rather than know explicitly. Of course, understanding differences in values for some variables is complicated by the fact that they were transformed. But, for instance, in Figure 3, it seems like compared to the active control, those in the social connectedness condition(s) asked, on average, between 1 and 1.5 more questions. Understanding that there was a statistically significant difference is one thing; understanding whether the difference is meaningful is another. Why does it matter that I ask 1 (or 1.5) more question over a 10 minute conversation? Does that really stand to make a difference?

Thank you for raising this point, we have now added the % increase based on the exponentiated coefficient for the condition effect, as the outcome was log-transformed (page 22): “Supporting our hypothesis, participants randomized to connect with strangers and acquaintances, compared

to an active control group, asked more follow-up questions during the structured conversation ($\beta = 0.15$, $b = 0.21$, 95% CI [0.04, 0.26], $p = .009$), corresponding to a 23% increase in the number of follow-up questions.”

We have also added raw descriptives directly in the results on Page 22: “On the raw (non-transformed) scale, the average frequency of follow up questions was 2.30 ($SD = 3.21$) in the social connectedness condition, and 1.45 ($SD = 2.21$) in the control condition.... On the raw (non-transformed) scale, the average frequency of verbal validation was 7.49 ($SD = 6.36$) in the social connectedness condition, and 6.90 ($SD = 5.45$) in the control condition.”

Based on our statistically significant results, it does indeed appear that asking an additional question can stand to make a difference. For someone that asks many questions, I would suspect an additional question may not be meaningful. But during an interaction in which a stranger asks a person two or more questions, compared to one or none, this additional question may make a meaningful impact in signaling interest to a partner.

For Study 2, we added a table to the supplemental material of descriptives by condition (Page 36: For descriptives of listening variables for the behavioral intervention group compared to the passive control group, see Supplemental Table S2.4.). However, it is slightly more difficult to interpret because we needed to compare per 30-second averages, given the variable length of conversation in this study.

Supplemental Table S2.4:

Table S2.4. Descriptives of listening behaviors by condition

30-sec Bin Average	Behavioral Intervention Group	Passive Control Group
	M (SD)	M (SD)
Global	1.48 (0.56)	1.47 (0.53)
Follow-up Questions	0.33 (0.26)	0.29 (0.24)
Verbal Validation	1.13 (0.75)	1.04 (0.64)

Lastly, and related to both verbal markers of listening (questions and validation), there is a difference between "asking a question" and the kind of question (or "validating emotion" and doing so explicitly vs. implicitly). Constructs like verbal person centeredness can help unpack what it means to do validation more/less well, and there are similar models of what constitute "good" questions. Perhaps the authors might provide some insight in the Discussion around how listeners might engage in some of these behaviors by looking at the transcripts and inferring the kinds of rhetorical strategies that could be potentially useful to train (so that we create

particularly "high" levels of "high-quality" listening).

Thank you for this suggestion. After reviewing the transcripts, we added a brief example of what high-quality listeners appear to do in conversation. This has been added to the discussion on page 39-40: “Notably, examination of the conversation transcripts revealed that individual’s rated high in quality listening often recalled specific details shared by their partners and referenced earlier content to progress the conversation with follow-up questions (e.g. “What were you thinking when the professor made that comment?”, “What happened after you told her what was on your mind?”).”

Reviewer #2 (Remarks to the Author):

Thank you for the opportunity to review the manuscript “High-Quality Listening Behaviors Linked to Social Connection Between Strangers: Implications 5 for Optimizing Behavioral Interventions”. While this manuscript touches on an important topic and has interesting findings, I see some major issues that need to be addressed before I recommend publication. I therefore recommend major revisions.

Abstract:

-The authors state that they test theory-driven behavioral and subjective markers of social connection, but from the main text I understood that these were also at least partially empirically proven in previous studies.

We use the term theory-driven to emphasize that our hypotheses are based in theory, we do not mean to suggest they are not empirically supported. To avoid confusion, we omitted “theory-driven” from our description in the abstract.

-The social connectedness interventions can be introduced more.

We can appreciate that details of the interventions are vague in the abstract. Given the word count, we decided to keep the information as is so as not to omit other information we believe is more essential to highlight.

-It is currently not clear that “asked their partner more follow-up questions” effectively means shows more listening behaviors.

To add greater clarity, we added “(i.e., displayed high-quality listening behavior)” following this phrase.

Introduction:

-Nice opening to emphasize the importance of interactions with strangers.

Thank you for the positive feedback on our opening.

-Line 73: The authors state that “people need simple, actionable strategies to improve connection in their daily interactions”, but is it not clear how you come to this conclusion based on the previous paragraph, and references are missing.

Thank you, we agree this transition could be improved and the claim better supported. We have now added (Page 3): “Despite the established positive effects of interacting with strangers, people often report avoiding these interactions alongside overblown fears regarding their own lack of conversational skills or enjoyment (Sandstrom & Boothby, 2021). These findings, in conjunction with the ongoing “loneliness epidemic” (Office of the Surgeon General OSG, 2023), suggests people may benefit from simple, actionable strategies to improve connection in their daily interactions, yet evidence-based behavioral recommendations are limited.”

-Line 96 – 102: Backchannel signals are also verbal, so what is meant by the explicit verbal expressions? Maybe an example could help here.

We have improved the clarity of this description on Page 4 (underlined text is the new addition): “Amongst the varied behavioral cues a listener may display, explicit verbal expressions (e.g., paraphrasing the speaker, expressing empathy) have been argued to be the strongest signal that conveys high-quality listening to a speaker (Collins, 2022). This is because backchannel responses (e.g., short verbal responses such as “yeah”, “uh huh”) and/or nonverbal listening cues (e.g., nodding)…”

-Line 102 - 105: “Between strangers, explicit verbal indicators may be especially useful for building connection, by helping interactants better anticipate one another’s thoughts and intentions, reducing uncertainty, signaling positive intention, and establishing interpersonal trust when there were no prior opportunities to do so.” This sentence needs more explanation and substantiation with references: I did not previously read about the uncertainty reduction and trust fostering nature of listening and why that is the case.

I have reworded this sentence for greater clarity and to keep consistent with arguments and evidence previously presented (On page 5): “Given the heightened uncertainty characteristic of interactions between unfamiliar partners, explicit verbal indicators may be especially useful for building connection by helping interactants better anticipate one another’s thoughts and positive intentions.” This sentence is echoing and building off the previous paragraph that outlines high quality listening theory, and the role of positive intentions.

-Line 179: The social connectedness interventions, later connection-based interventions, appear out of the blue here. What are they? This needs to be explained.

Thank you for raising the opportunity to improve our introduction of the interventions. We have now added a greater introduction to the rationale for including social connectedness interventions, and generally what the interventions consisted of (Page 8): “In the present study, we draw on two secondary datasets, each of which consisted of an intervention aimed at improving quality social connection (i.e., positivity resonance). Given theory that high-quality listening precedes social connection (Zhou & Fredrickson, 2023), we had the opportunity to additionally examine whether high-quality listening behaviors during post-intervention interactions emerge as a potential mechanism through which each intervention successfully promoted social connection. Each study previously reported evidence that an intervention that encouraged participants to have more high-quality connections (compared to controls) was successful. Specifically, the intervention of Study 1 successfully raised markers of social

connection during an in-lab interaction 48-hours later (West et al., 2024) and, that of Study 2 improved self-reported positivity resonance in daily interactions across 35 days (Zhou et al., 2021).

-I feel there are some inconsistencies in the argumentation: first the authors mention that there has been a lot of listening research looking at indicators of social connection, but that this is mostly based on experimental data where listening is manipulated, so less naturalistic, but later (line 184) you say that most research is based on self-report perceptions of listening, which sounds rather naturalistic.

Thank you for raising this point of confusion. We are referring to naturalistic *behavior*. That is, behaviors that are observed and that have not been trained or manipulated by the experimenter (as in manipulating listening behaviors for experimental purposes). As you also later note, we understand naturalistic may not be the best way to describe our research as participants are in lab settings, so these are not unambiguously “real-world” observations. We clarify on page 6 that we are referring to interactions that do not have manipulated behaviors or self-reports of perceived listening. On page 6, we added “observed” to further highlight this distinction (the underlined text reflects new additions) : “However, limited empirical evidence directly supports these posited links, and of the existing empirical evidence, none has investigated listening behaviors that are merely observed and coded systematically (i.e., in comparison to manipulated or captured via reported perceptions) or behavioral markers of social connection.”

Study 1:

I have a couple of questions concerning the methods of Study 1 (and 2) that need to be clarified in the manuscript:

-I understand that “Sample size was determined based on a power analysis for the original aims of the grant-supported study”. But I would still like to know what your power was for the current study. This also applies to Study 2.

We have now included sensitivity power analyses for both studies.

In Study 1 (Page 11): “A sensitivity power analysis was conducted in G*power 3.1 (Faul et al., 2009) for a linear multiple regression with 1 primary predictor and 4 covariates ($N = 298$, $\alpha = .05$ (two-tailed), power = .80). Primary models are powered to detect a minimum effect size of $f^2 = 0.026$ (small).”

In Study 2 (Page 27): “A sensitivity power analysis was conducted in G*power 3.1 (Faul et al., 2009) for a linear multiple regression with 1 primary predictor and 4 covariates ($N = 348$, $\alpha = .05$ (two-tailed), power = .80). Primary models are powered to detect a minimum effect size of $f^2 = 0.023$ (small).”

-Did students sign informed consent for Study 1 and 2? This should be explicitly stated in the manuscript. Also, was there a cover story for participants and what was this?

We have added to both study procedures that participants provided informed consent prior to participation (Page 11 and 28). In Study 1, participants are told that the study is to test the effect of technology on health behavior goals. There was not a cover story pertaining to why participants were asked to carry out the semi-structured conversation with another student beyond our intended interest in health behavior. In Study 2, the cover story for the social interaction can be found under “Procedure”.

- Why was the diaphragmatic breathing intervention chosen as active control group?
- What was the short educational video (watched before interacting with Ellie) about?
- Did students keep to the assignment they were given to practice in the intervening 48 hours? I think this is an important aspect of both Study 1 and 2 (with its 35 days interval) since that is the social connectedness intervention the authors aim to test.

We have added details regarding the intervention videos to the procedure (Page 12): “This active control group was selected to be a simple non-social health behavior that participants could frequently enact across 24-hours, to parallel the cognitive and motivational effort of the treatment group (West et al., 2024). In both conditions, the intervention consisted of a three-minute educational video followed by a conversation with a virtual human avatar called “Ellie”. Participants randomized to the social connectedness condition watched a video about the benefits of positive connections with strangers and acquaintances (<https://www.youtube.com/watch?v=tXTj4mDON7k>) and encouraged participants to be more attentive and open to connection with strangers and acquaintances. Participants randomized to the diaphragmatic breathing condition watched a video about the benefits of using proper breathing techniques (https://www.youtube.com/watch?v=27_Z-zaFb88) and encouraged participants to focus on allowing their breaths to expand from their back and belly.” We also included the full Ellie (virtual human) script in the supplemental material: “In both conditions, Ellie asked participants to create behavioral implementation intentions for their assigned behavioral goal (i.e., If-Then plans, Gollwitzer & Sheeran, 2006; see Supplemental Section II for full Ellie script).”

We also now add to the introduction evidence that both interventions did successfully improve social connection, suggesting participants were carrying out behavioral goals (Page 8): “Each study previously reported evidence that an intervention that encouraged participants to have more high-quality connections (compared to controls) was successful. Specifically, the intervention of Study 1 successfully raised markers of social connection during an in-lab interaction 48-hours later (West et al., 2024) and, that of Study 2 improved self-reported positivity resonance in daily interactions across 35 days (Zhou et al., 2021).”

- Why did expressing simple agreement (e.g., “me too”) not count as verbal validation? (line 278-279).

When developing the coding scheme and during early training, we made the decision to not count “me too” as verbal validation. Expressing agreement in this way was common and we

interpreted it as a plausible attempt to shift the conversation back to oneself rather than acknowledging or reflecting the speaker's statement. Thus, simple agreement on its own did not meet our criteria for validation of the speaker. We briefly elaborated on this decision (underlined text reflects the new addition) on page 13: "This did not include vocal backchannel indicators (e.g., yeah, mmmm) or turning the conversation back to the self, including expressing simple agreement (e.g., "me too"), rather than first acknowledging or reflecting the speaker's statement."

-Line 285: How did the authors decide when reliability was met in the coder training?

-Line 286: The authors state that to assess reliability, all coders coded 20% of the videos, but with 5 coders, this comes down to 100% of the videos, so 0 overlap right? I think this is a phrasing issue, since the authors probably mean to say that a set of 20% of the videos were coded by all 5 coders.

Thank you for catching this – you are correct and we have rephrased this sentence to read "a set of 20% of the videos were coded by all 5 coders" (Page 14). We additionally clarified how reliability was assessed on Page 14: "Once reliability was met (i.e., ICC > .80) in training..."

-Line 298: The authors state that "There were 16 "unusable transcripts with major errors", primarily due to poor audio quality. It is unclear what the authors did with these transcripts. Were they excluded from the analysis, and also from the listening coding that was described earlier? I ask the authors make this clear and, if it applies to all qualitative analysis, mention it earlier.

Thank you for raising this lack of clarity. We have added that these cases were removed solely from analyses that used conversational response time. Transcripts were generated using auto-transcribers, not humans. If the audio was distorted or muffled, the auto-transcriber was unable to pick up speech as well. We did not, however, have this same issue when using humans to code listening behaviors directly from the video records. Upon reassessing our analytic code, we also observed that in our analyzed sample for conversational response time, only 12 cases were omitted. I have updated this information and added on Page 14-15: "Research assistants manually checked transcriptions for completeness and accuracy and flagged unusable files with major errors, primarily due to poor audio quality that resulted in missing or unreliable auto-transcription. Of the data used in the present study, 12 transcript files were flagged as unusable and were not included in response time analyses."

-I think it is not entirely fair to treat the confederate's and later experimenter's (as the interaction partner) experiences and behaviors as 'naturalistic', since both of them did know the purpose of the study, I assume. Moreover, and more importantly, they were partly instructed to behave in a certain way and I would assume that they developed some behavioral patterns during the many interactions they had. The authors do reflect on this in the Discussion section, which I appreciate, but it is not entirely clear how they "controlled for confederate/experimenter effects" in their primary models. Can they elaborate on that? I would also like a mention of this limitation earlier on, when writing about the study designs.

As noted earlier, we removed the word “naturalistic” as a descriptor of the staged small talk. In response to Reviewer 1, we have also expanded on our use of confederates and how we controlled for the effect of individual differences (See Pages 2-3 of this response letter). We have also added this limitation to the end of the introduction on page 9: “For logistical and experimental control, albeit at the cost of potential practice effects, each participant’s partner was either a trained confederate (Study 1) or experimenter (Study 2), in each study blinded to participant’s condition.”

-I would like to see some descriptives, either in the Results or in the Methods of the average positivity resonance as well as the behavioral coding, to get a general impression of the dataset. For the rest, as far as I can judge, the analyses appear sound and the results clear.

Thank you for this suggestion, we have now added descriptives to results for both studies at the opening of each respective Results section. As noted to Reviewer 1, we also added per condition listening variable descriptives (See page 4-5 of this response letter).

On page 18: “Descriptives of study variables are presented in Table 1.”

Table 1. Study 1 Descriptive statistics for primary study variables

Variable	M (SD)	Scale
1. Avg Gap Length - Participant	1.01 (0.79)	Seconds
2. Avg Gap Length - Partner	0.86 (0.65)	Seconds
3. Participant-reported PosRes	7.87 (1.53)	0-10
4. Partner-reported PosRes	7.91 (1.67)	0-10
Behavioral Codes per 30s bin		
5. BIPR	0.39 (0.32)	0-2
6. Verbal Validation	0.72 (0.59)	Avg frequency
7. Follow-up Questions	0.19 (0.28)	Avg frequency

Note. *M* and *SD* are used to represent mean and standard deviation, respectively. PosRes = positivity resonance. BIPR = behavioral indicators of positivity resonance. We report per-bin averages (by dividing by 10) to allow for comparison to Study 2 descriptives. Avg frequency refers to the average count frequency across coders per bin.

On Page 32: “Descriptives for study variables are presented in Table 3.”

Table 3. Study 2 Descriptive statistics for primary study variables

Variable	M (SD)	Scale
1. Avg Gap Length - Participant	0.43 (0.30)	Seconds
2. Avg Gap Length - Partner	0.44 (0.30)	Seconds
3. Partner-reported PosRes	8.08 (1.71)	0-10

Behavioral Codes per 30s Bin

4. BIPR	0.58 (0.33)	0-2
5. Global Listening	1 (0.55)	0-4
6. Verbal Validation	1.10 (0.72)	Count
7. Follow-up Questions	0.32 (0.26)	Count

Note. *M* and *SD* are used to represent mean and standard deviation, respectively. PosRes = positivity resonance. BIPR = behavioral indicators of positivity resonance.

Additionally, to limit the number of tables, we removed the correlation table under Study 2 results, and instead wrote out the correlation statistics for the Validation Hypothesis on Page 32:

“In support of our Validation Hypothesis, we found that the new global listening behavioral coding scheme was positively correlated with both the log-transformed frequency of follow-up questions ($r(344) = 0.54, p < .001$) and verbal validation ($r(344) = 0.67, p < .001$). Verbal validation and follow up questions were also positively correlated ($r(345) = 0.32, p < .001$).”

Study 2:

-See comments under Study 1.

-I ask the authors to indicate the number of participants per treatment (and control) group. And to reflect on the fact that the groups they later compare (all treatment groups vs. the one control group) probably are very different in size.

We now indicate the sample size for each group, and have also added to the discussion a brief note that given our combined active intervention group for analysis, our comparison group is roughly a third the size.

On Page 41-42, in the discussion: “Because all three active conditions showed improved self-reported positivity resonance over the course of the intervention compared to the passive control group (Zhou et al., 2022), we combined active conditions to test for effects on listening. While our control group was roughly a third the size of the combined active group in analyses, this decision was intended to maximize statistical power to detect treatment effects. However, the resulting unequal group sizes may have reduced the precision of estimates for the control group, particularly in detecting small effects.”

-I am not sure but isn't it rather unusual to refer to another paper (about the same dataset) for more information about the study setup? I would assume the current manuscript should be a stand-alone paper and I would therefore recommend the authors to include an appendix or online materials with these same details.

Thank you for raising this, we included further details on the interventions directly in the Methods sections, particularly in Study 1 as noted earlier in this response letter. There were not additional details for Study 2 that were not already presented in our current manuscript, so we also changed how we referenced the original manuscripts, from “for details see...” to “for the original report, see....”.

-I would also recommend rephrasing “naturalistic context” to something more fitting, since, as I indicated before, I find this setting with an experimenter as interaction partner not very naturalistic. Otherwise, the authors should clearly explain how this context is naturalistic.

We have omitted the use of “naturalistic” to describe our study design.

-I would like to know why only the experimenter reported on positivity resonance and the participants did not (since this is an important limitation).

Thank you for raising this point. We now clarify in our measures section why self-reported positivity resonance was not collected from participants. We also wish to note that the current manuscript is a secondary analysis, so study design decisions were not made with the present aims in mind.

On Page 30, under measures: “Due to the mild deception used to stage an opportunity for small talk with the experimenter, self-reported positivity resonance was not collected from participants.”

-I am not sure whether I understand why the non-verbal listening cues were now included after all, since the authors clearly defended their focus on verbal listening in the Introduction. Can the authors clarify this?

Thank you for raising this opportunity for greater clarity. We did not intend to suggest that *only* verbal cues should be focused on, rather simply that they may be particularly important or may to a greater degree drive perceptions of listening for speakers. We did initially intend to assess an independent dimension of nonverbal cues in Study 1, yet did not proceed with coding due to difficulty achieving reliability. We added this footnote on Page 10 (see below). In Study 2, we aimed to build on the evidence provided by verbal indicators in Study 1 to see whether a global measure that incorporated nonverbal behaviors *in addition* to verbal expressions would produce similar results. Having one global measure may also be more efficient for other researchers to employ, rather than coding for multiple verbal indicators. Although we stand by our argument that verbal expressions may be the strongest honest signal of high-quality listening, we also make the case for taking a holistic view: that the presence of relevant nonverbal cues alongside verbal expressions may further amplify this effect. The following additions have been made to the manuscript:

On Page 26, Study 2 Introduction: “Furthermore, having established in Study 1 that verbal indicators of high-quality listening are linked to social connection, we sought to expand on these measures to also test a holistic measure of global listening behaviors that considers non-verbal behaviors together with explicit verbal expressions. Although explicit verbal expressions may provide the strongest “honest signal” of high-quality listening, the extent to which nonverbal behaviors are present may serve to further amplify that signal. In Study 2 ($N = 348$), raising the bar to a brief, small talk context, we tested whether high-quality listening behaviors, measured both as verbal indicators (as in Study 1) and as a global evaluation of verbal and non-verbal

indicators, are linked to high-quality social connection in a brief conversation with an unfamiliar partner.”

On page 10, a footnote in the Study 1 introduction:

¹ We originally planned to assess a third cue, non-verbal indicators of listening. Due to difficulty achieving reliability across coders (i.e., ICC > .80), we dropped this indicator early in coder training to prioritize verbal indicators.

Discussion:

-Some of the findings described are not clear for readers that skip or skim the Results section. I recommend to clarify this by repeating the main findings instead of simply stating that they were “replicated” as is done now (lines 712-714).

Thank you for this suggestion. We have now repeated the pattern of findings in the introduction to the general discussion. On Page 37: “Specifically, both verbal validation and follow-up questions were linked to participant and partner conversational response time and behaviorally coded and partner-reported positivity resonance, but not participant’s self-reported positivity resonance. When testing this same hypothesis in a “small talk” conversation that better approximates real-world interactions with strangers (Study 2), we found the same pattern of findings when assessing global listening behaviors (i.e., verbal and non-verbal indicators in combination). Specifically, global listening behaviors were associated with all tested markers of social connection (i.e., participant and partner conversational response time and behaviorally coded and partner-reported positivity resonance; participant-reported positivity resonance was not measured in Study 2). Verbal validation followed the same pattern, except was not linked to partner conversational response times. Finally, in Study 2, unlike Study 1, participants’ follow-up questions were only associated with partner response time (not the questioner’s own response time or positivity resonance).”

-I miss speculation for why the authors find an effect for the intervention in Study 1 and not in Study 2 (maybe the different time duration can be a factor?).

We speculate on this in toward the end of the discussion on Page 41-42. We agree that the time duration could be a factor. We also discussed how the quality of the intervention was enhanced in Study 1 from Study 2 by incorporating several additional features known to facilitate behavior change (i.e., implementation intentions).

Minor points:

-Prevent abbreviations, in line with APA guidelines, so write out things like “needn’t” (line 72).
-I recommend a thorough read through because there are some writing mistakes and missing words, e.g., line 198: “listening behaviors are raised by an intervention designed improve social connection”. In this sentence the word “to” is missing, also I am wondering whether “raised” which is used more in this context is the right term.

Thank you for these minor points. We have proofread the manuscript and removed abbreviations.

Author Responses to Reviewer

Thank you for your careful consideration of our manuscript, now entitled “High-Quality Listening Behaviors Linked to Social Connection Between Strangers:” (COMMPSYCHOL-25-0180). Below you will find our detailed responses, with the complete and verbatim text from the reviewer presented in black, alongside our responses in blue text. All page numbers mentioned refer to the revised version of the manuscript. We have also highlighted changes directly in the manuscript, as requested. New changes have been highlighted in green, while we have left changes from the previous revision in blue.

REVIEWER EXPERTISE:

Reviewer #1: social connection, behavioral coding

REVIEWER REPORTS:

Reviewer #1 (Remarks to the Author):

I've completed my review of "High-quality listening behaviors linked to social connection between strangers." Overall, I believe the manuscript stands to inform us about "simple, actionable strategies to improve connection in their daily interactions" (p. 3). As a listening scholar, I appreciate the attention this article might bring to how we conceptualize, measure, and teach such an important (yet often underappreciated) life skill.

Although in our age of electronic articles "journal space" is typically not an issue, I would still encourage the authors to find places to streamline arguments and delete text. For instance, information on page 8 ("In the present study...") is largely repeated on page 12 ("...we drew from two large, archival datasets ..."). My recommendation is to reduce the 9-paragraph introduction to 4-5 paragraphs. Within that introduction, when the authors speculate about reasons that "explicit verbal expressions" operate as "stronger" indicators of "high-quality listening" they might benefit from the logic used by Herb Clark in his 1996 book, *Using Language*. In that book, Clark discusses "evidence of understand" in his model of discourse understanding (and how "grounding" happens). In that discussion, he outlines various types of evidence that listeners use to signal they have understood the speaker "well enough for current purposes." When listeners use "explicit verbal expressions" they are more directly signaling their understanding (versus when they use backchannels or other short utterances). So, while it is true that "backchannel responses ... can be deceptively used when people merely pretend to listen" that is not the only explanation (and in my opinion not the best explanation) for why these cues are weaker signals that one is listening.

Thank you for the suggestion to streamline our introduction. We have especially focused on streamlining the latter part of the introduction in which we describe the study, resulting in cutting one repetitive paragraph. While we did not remove any other

full paragraphs, we did tighten the introduction by removing repetitive sentences or information not essential to our argument. Overall, whereas the previous introduction was 2,148 words, the current introduction is now 1,596 words. We also appreciate the excellent suggestion to reference the logic put forward by Herb Clark on the levels of evidence of understanding. We have included this explanation as to why verbal indicators are strongest, in addition keeping theorizing as to why weaker signals can fall short. The underlined text is the new addition and reworking of to this paragraph.

Page 4: “Amongst the varied behavioral cues a listener may display, explicit verbal expressions (e.g., paraphrasing the speaker, expressing empathy) have been argued to be the strongest signal that conveys high-quality listening to a speaker, as they can provide direct evidence of having understood the speaker (Clark & Schaefer, 1989; Collins, 2022). In contrast, weaker evidence, such as backchannel responses (e.g., short verbal responses such as “yeah”, “uh huh”) and/or nonverbal listening cues (e.g., nodding) can be deceptively used when people merely pretend to listen (Collins, 2022).””

In both studies, I recommend the authors report the exact reliability estimates - for instance in the second footnote on page 10 (instead of ICC > .80, report the actual reliability estimate obtained) and on page 14 (once reliability was met, ICC = ??). In that same paragraph on page 14, I am not clear as to whether the "same set of videos" was coded "each week" (that is whether each coder coded videos 1, 2, and 3 or whether one coder coded 1-3 and the other coded 4-6). I also was confused by how an "average frequency of follow-up questions and verbal validation across coders were used for analysis" if "only 20% of the videos were coded by all 5 coders." How is there an average if 80% of the videos were coded by a single coder?

Thank you for the opportunity to add clarity on our reliability reporting. We want to clarify that we do provide the exact reliability estimates for the sets of codes used in any analyses. We simply were noting the criteria for assessing reliability during training. However, we have now added training reliabilities for nonverbals at the end of week 2 that were not originally included in the manuscript, as we did not continue training nor used those codes in analyses. We moved this footnote into the supplemental material and included a note in the manuscript to reference supplemental material for further context on our decision not to pursue nonverbal coding for Study 1.

Page 9: “(See Supplemental Section I for a discussion of why we elected to forego pre-registered coding for nonverbal listening indicators).”

Supplemental Section I: “Although we had pre-registered additionally assessing nonverbal indicators, attempts to code purely nonverbal cues of listening were aborted early in coding training (week 2 of 6) due to combined concerns about validity as well as reliability. Although reliability did not yet reach our threshold of ICC >.80 (nonverbal indicators average ICC of .75 and single-coder ICC of .37), our decision was ultimately driven by a loss of confidence in the nonverbal cues coded as valid measures of high-quality listening when considered in isolation.

We therefore narrowed our focus to verbal indicators for Study 1, which have been argued to be the strongest behavioral signals of high-quality listening.”

Regarding the coding, we state in that paragraph that two people coded each video, however, for 20% of the videos, all five coders coded the videos to assess reliability across coders. The remaining 80% of videos only had two coders per video. There was no video coded by a single coder.

I am particularly interested in the data presented as Figure 3. There appears a great deal of variability in the number of questions (as well as the amount of verbal validation), and yet experimental condition only explained a small percent of that variance (helpful for me to look at d values which were all less than .5, corresponding to an r-squared of less than .05). What explains the remaining variability - it does not seem that this study tracked the extent to which people participated in the intervention; but perhaps practicing everyday vs. not much practice may have something to do with that variability? On page 26 (Study 2), the authors mention "daily reporting" - are there data that could answer this?

We appreciate the reviewer’s insightful observation regarding the variability in listening behaviors. Indeed, as noted, the intervention condition explained only a small proportion of variance, and it is likely that a number of other predictors may explain additional variance (e.g., personality characteristics, motivation, relational dynamics). The purpose of this analysis was to test whether the intervention increased listening behaviors, not to identify the sources of variability across individuals for engaging in listening behaviors. Furthermore, this study was only a 48-hour intervention, so there was not daily reporting (that was only in Study 2, which did not find intervention effects). We agree that understanding additional sources of variance would be an important avenue for future research and now note this in our discussion.

On page 34-35: “...suggesting that additional processes beyond listening behaviors also contribute to the intervention’s impact on improved social connection (e.g., personality, relational dynamics, motivation).”

Those distributions are also interesting insofar as the mean # questions was quite low compared to verbal validation. Is this a function of presence/absence of these variables (conceptual question) or how they are operationalized (measurement question)? There seem to be more ways to express verbal validation but only one way to ask a question - and as the literature on verbal response modes (largely in the context of therapy) suggests, the nature of a question can be grammatical or pragmatic/use-based (meaning I can "ask a question" even if my wording is not phrased grammatically as a question; e.g., I could say, You're leaving already with rising intonation for instance). In any event, the fact there is so much variability in behaviors left to be explained seems in need of an explanation. Overall, these are TINY effects (when they occur), and the effects that are there are not fully consistent (especially between the two studies). It is, thus, not 100% accurate to claim, for instance, that "people may intuitively ask their

partner more follow-up questions to increase connection" - verbal validation was much more common (hence more intuitive); plus the increase was very very very small.

We thank the reviewer for their thoughtful comments. We understand that there is unexplained variability, and the effects are small. Even so, we find evidence that the intervention had a statistically significant effect on follow-up questions. We underscore that the intervention only had an effect on follow up questions, and not verbal validation. Thus, in our study specific discussion, we speculate that people who were given instructions to connect may have done so by asking follow up questions, as that is what we have evidence for as being influenced by the intervention. Whether or not verbal validation was more common, we did not find that it was influenced by the intervention. In our study-specific discussion, this speculation is tied directly to the intervention finding (i.e., when people were given instruction to connect), and not how people intuitively try to connect in general.

The effects on follow-up questions are inconsistent across studies, and the two studies were different interventions and used different conversational contexts. Our discussion spends multiple paragraphs discussing plausible reasons for inconsistent effects due to distinct aspects of the intervention design. To increase transparency, we have added multiple notes that effect sizes were small when discussing the findings

Directly in results reporting, page 26: "This was a small but statistically significant effect"

And in the general discussion, page 34: "Although this might suggest that the intervention's success in increasing connection quality may be partly explained by participant's asking more follow-up questions, we hasten to underscore that effect sizes were small...."

I am also a bit concerned that the inclusion of "both verbal and non-verbal components" of listening in Study 2 was largely driven by the fact that the coding of non-verbal behaviors in Study 1 failed. There are several places between the end of Study 1 and the beginning of Study 2 that the authors present this as an advancement of Study 2, but in reality it was something they thought of already for Study 1 (but it just did not work out).

Thank you for raising this opportunity to further clarify the decision to not proceed with nonverbals in study 1. As we have transparently stated, during training, we decided to drop the nonverbals during week 2 of 6 weeks of training. We'd like to emphasize this decision was made early in the training period, well before formal behavioral coding for Study 1 videos created the data analyzed here. Although we might have decided to alter the code or continue training for nonverbal cues (as you may note, the reliabilities were not that poor), our concerns were both conceptual and statistical. When applying the nonverbal coding scheme and discussing disagreements during reliability meetings we lacked confidence that nonverbals reflected listening when observed in isolation. Thus, we decided to drop the nonverbal coding to focus first on verbal indicators. To state it

plainly, we didn't "fail" in our coding of nonverbal cues for listening that we had preregistered. Rather, early in the training phase, once we viewed and discussed more training videos, we identified ambiguities in the observed behavior that led us to question the validity of using these cues to index listening and chose not to pursue this form of coding further. Then, for our next Study 2 we changed our coding approach to create a more streamlined global coding system that holistically incorporated both verbal and nonverbal components. We underscore that global coding was not our original plan for study 1. It was newly developed for Study 2. Thus, we took what we learned from the first study to advance our coding scheme in a second study. We hope that our current revision will prevent any similar misunderstanding of our coding processes across studies and in doing so avert any concern. We further clarified our reasoning in our supplemental material, as noted in our earlier response above, but we also re-paste below

Supplemental Section I: "Although we had pre-registered additionally assessing nonverbal indicators, attempts to code purely nonverbal cues of listening were aborted early in coding training (week 2 of 6) due to combined concerns about validity as well as reliability. Although reliability did not yet reach our threshold of ICC $>.80$ (nonverbal indicators average ICC of $.75$ and single-coder ICC of $.37$), our decision was ultimately driven by a loss of confidence in the nonverbal cues coded as valid measures of high-quality listening when considered in isolation. We therefore narrowed our focus to verbal indicators for Study 1, which have been argued to be the strongest behavioral signals of high-quality listening."

Like the introduction, I also think the Discussion is a bit overwritten. The authors might also usefully combine many aspects of Study 1 and Study 2 to try and reduce the length of the manuscript. They both answer a similar set of questions, and the methods are quite similar.

Thank you for this feedback. We have combined studies according to editorial requests and streamlined the discussion. Within results, we also greatly reduced (for Study 1) or removed study specific discussions (for Study 2) given this new format. Whereas our previous general discussion was 2,578 words, in the current revision it is 1,869.